# Cardio-mmFlow: A Gaussian-Prior-Free Physics-Informed Flow Matching Framework for Electrocardiogram to mmWave Radar Synthesis

Ziyang Liu [1]  Ruiqiang Xiao [1]  Chang Huang [1]  Kieren Yu [1]  Siyuan He [1]  Kaishun Wu [1]

## Abstract

Continuous ECG monitoring is clinically valuable, but scaling it beyond electrodes to comfortable long-term use motivates contactless mmWave sensing. In practice, mmWave-to-ECG reconstruction is severely constrained by the scarcity of high-quality synchronized recordings. Therefore, we propose **Cardio-mmFlow**, a Gaussian-prior-free physics-informed flow matching framework that synthesizes mmWave radar signals from clinical ECG. It learns a direct transport trajectory between the latent manifolds of ECG and radar. Considering subject-dependent propagation differences, we incorporate a simplified mass–spring–damper inspired modulation and inject it into the flow dynamics via feature-wise linear modulation for personalization. Extensive experiments show that our system generates high fidelity radar data in both signal and latent domains. It significantly improves zero-shot downstream mmWave to ECG task, and enable Atrial Fibrillation classification with synthetic data. Further analyses evaluate the model interpretability.

## 1. Introduction

Cardiovascular diseases (CVDs) remain the leading cause of mortality worldwide, accounting for millions of deaths annually and imposing a substantial public health and economic burden (Chong et al., 2025). Continuous monitoring of cardiac activity is therefore vital for early diagnosis and long-term health management (Jaspan et al., 2024; Hughes et al., 2023). Electrocardiography (ECG) remains the clinical gold standard for characterizing cardiac rhythm and related pathologies. This need is especially pronounced during sleep, which occupies roughly one-third of human life and is a period when nocturnal arrhythmias and other cardiac events commonly occur. Despite this clinical importance, traditional ECG acquisition is not friendly in sleeping scenario. ECG need skin-contact electrodes which are prone to detachment (Joutsen et al., 2024), may cause skin irritation (Fontana et al., 2019), and typically require cumbersome setup. Even wearable or patch-based ECG systems still face practical constraints, discomfort during extended use reduces tolerability and adherence (Hughes et al., 2023). Moreover, regular electrode replacement and vulnerability to everyday conditions such as perspiration and body movements often lead to unstable recordings (Li et al., 2021). Collectively, these issues hinder truly continuous and user-friendly sleeping cardiac monitoring.

With the advancement of Radio Frequency (RF) sensing technology, researchers have explored RF signals as a way to realize contactless cardiac monitoring. Early studies relied on commodity WiFi (Wang et al., 2020; Zhang et al., 2020) and Ultra-Wideband (UWB) signals (Hu & Jin, 2016; Lee et al., 2018) to detect heartbeat-induced variations in the wireless channel. Subsequent work introduced high-resolution millimeter-Wave (mmWave) radar to capture subtle chest-wall micro-vibrations, which has been widely utilized for heart rate and heart rate variability (HRV) estimation (Wu et al., 2023b; Wang et al., 2021; Yang et al., 2016). Driven by the powerful representation capabilities of deep learning models, recent research has moved beyond simple vital sign to assess complex cardiac conditions (Yuan et al., 2025; Zhao et al., 2024b). Several studies have demonstrated the feasibility of reconstructing ECG waveforms from these mechanical vibration patterns, effectively bridging the gap between contactless mechanical sensing and clinical electro-physiological standards (Wang et al., 2023; Chen et al., 2022; Zhang et al., 2025b; Xu et al., 2021; Wu et al., 2023a; Zhao et al., 2024a).

Despite recent progress in contactless cardiac monitoring, deploying contactless ECG methods in unconstrained home environments remains challenging. Most existing systems are trained on limited and lab-collected paired datasets. Their performance often degrades sharply on unseen individuals, which revealing a persistent cross-subject generalization gap. This gap is largely rooted in the difficulty of col-

---

[1] The Hong Kong University of Science and Technology (Guangzhou), Guangzhou, China. Correspondence to: Kaishun Wu <wuks@hkust-gz.edu.cn>.

*Proceedings of the 43rd International Conference on Machine Learning*, Seoul, South Korea. PMLR 306, 2026. Copyright 2026 by the author(s).

lecting and sharing high-quality synchronized radar–ECG data. Building such datasets requires strict synchronized clinical-grade ECG ground truth, a high-barrier protocol that restricts cohort scale and demographic coverage. Data release is further impeded by privacy regulations and NDAs, resulting in fragmented "data silos" (Saberi et al., 2025). Moreover, many clinically important cardiac events (e.g., paroxysmal arrhythmias) are intermittent and may not occur during standard recording sessions (Zhao et al., 2024b), making real datasets not only small but also event-sparse.

In this work, we focus on **mmWave radar** as the contactless sensing front-end, as its phase measurements are highly sensitive to sub-millimeter chest micro-motions. Collecting a small amount of calibration data from each new user and fine-tuning the model is a practical way to mitigate cross-subject distribution shift. However, this strategy still requires synchronized radar–ECG recordings for each deployment, which introduces substantial labor cost and makes large-scale extension difficult. Motivated by data synthesis paradigms in the AI community, such as face reenactment (Thies et al., 2016), we explore a complementary direction: leveraging large-scale public ECG datasets to synthesize **diverse and personalized mmWave radar signals** as training supervision. We aims to reduce the reliance on extensive paired radar–ECG collection by generating radar-side supervision from widely available ECG resources.

To realize this task, we face two fundamental challenges. The **(i) Challenge one** is *Constructing a physiology-grounded generative trajectory.* Most generative frameworks synthesize signals by evolving from an unstructured Gaussian prior. Such noise initialization is suboptimal in our paired, beat-synchronous ECG→mmWave setting: ECG and radar segments are two measurements of the same cardiac cycle, so discarding the ECG-implied beat morphology and timing at initialization introduces unnecessary ambiguity. We therefore seek a direct latent transport that starts from the ECG latent codes and evolves them toward the radar latent manifold. The key challenge is to bridge the latent mismatch between ECG and radar and learn a direct transport trajectory that remains faithful to the physiology implied by ECG, rather than drifting into unrelated radar patterns. The **(ii) Challenge two** is *modeling subject-specific physical variations in signal propagation*. ECG provides a clinically interpretable rhythm template, whereas mmWave radar measures thoracic surface micro-displacements. So radar observes a propagation- and geometry-modulated manifestation of cardiac activity rather than the electrical source itself (Chen et al., 2024). Subject-dependent factors (e.g., anatomy and tissue mechanical properties) alter how cardiac-induced vibrations propagate to the skin, yielding different attenuation patterns and resulting phase-modulation signatures in the measured mmWave signals (Sandler et al., 2023). Therefore, faithful synthesis requires subject-conditioned

propagation information to generate individualized attenuation patterns and phase-modulation responses.

To this end, we introduce **Cardio-mmFlow**, a physics-informed generative framework for ECG→mmWave synthesis. Cardio-mmFlow departs from conventional noise initialization by transporting structured ECG latents to the radar latent manifold via flow matching. Build upon two pre-trained VAEs define modality-specific manifolds, then we couple distribution alignment and trajectory learning in latent space: a Wasserstein objective aligns cross-modal latent distributions, while a Gaussian-prior-free flow-matching model learns a direct transport path from ECG to mmWave radar. To model subject-specific propagation variability, we incorporate an simplified Mass-Spring-Damper (MSD) inspired feature-wise linear modulation (FiLM) driven by biometric metadata to induce personalized phase attenuation. Finally, we add a spectral fidelity loss to refine frequency characteristics, and adopt classifier-free guidance (CFG) to strengthen conditional adherence during generation.

The contributions of this work are as follows,

- We propose **Cardio-mmFlow**, a physics-inspired framework for ECG to mmWave synthesis. To the best of our knowledge, this is the first work to synthesize mmWave radar cardiac waveforms from public ECG datasets, filling a critical research gap in the contactless ECG community.
- We design a Gaussian-prior-free latent flow matching model that couples Wasserstein distance based latent alignment with trajectory learning, and incorporate an MSD-inspired FiLM to model subject-specific propagation effects with user metadata.
- Extensive experiments demonstrate that the synthesized mmWave signals preserve strong time–frequency fidelity and improve downstream cross-subject mmWave→ECG reconstruction under **zero-shot** settings. We further validate the generated signals through cardiac disease detection task and interpretability analysis, showing their potential to support downstream cardiac sensing applications.

## 2. Related Work

**Contactless ECG via Radar sensing.** mmWave radars detect sub-millimeter thoracic motion through phase shifts in the reflected waves. This enable the radar effectively sensing the cardiac mechanical events. When the chest surface moves by a small displacement $d(t)$, the reflected phase $\phi(t)$ changes according to (Singh et al., 2020)

$$\Delta\phi(t) = \frac{4\pi}{\lambda}d(t), \tag{1}$$

where $\lambda$ is the radar wavelength. The micro-displacements $d(t)$ that modulate RF phase, creating a signal that remains

time-locked to the ECG-induced electrical cycle.

The task maps thoracic micro-vibrations to cardiac electrical activity which similar to cross modality transformation. Early efforts like CardiacWave (Xu et al., 2021) pioneered physics-based scattering models, which were subsequently superseded by deep architectures such as mmECG (Chen et al., 2022) and RSSRnet (Wu et al., 2023a) to exploit high-dimensional RF signatures. More recent works have introduced generative and differential paradigms, including GAN-based RF-ECG (Wang et al., 2023), diffusion-based AirECG (Zhao et al., 2024a), RadarODE (Zhang et al., 2025b) and mmJEPA-ECG (Liu et al., 2026). While these models have steadily improved reconstruction fidelity, they remain fundamentally data-hungry, relying on subject-specific labeled datasets that are difficult to acquire in home-care settings.

**Synthetic data works in Radio Frequency (RF).** The scarcity of high-quality, large-scale training datasets has long limited progress in the wireless sensing community (Cai et al., 2020). Data synthesis has therefore become an important strategy, either by simulating physical propagation mechanisms (Deng et al., 2023) or by generating signals through deep generative models. This approach has shown clear benefits in several RF tasks, especially human activity recognition (HAR), where synthetic CSI or radar signals have been used to enhance robustness and improve generalization (Chen & Zhang, 2023; Gong et al., 2025; Chi et al., 2024; Zhang et al., 2022). However, most existing RF synthesis pipelines are tailored to macro-scale dynamics (e.g., HAR) and often rely on coarse time–frequency representations, which discard the fine temporal structure required to capture cardiac micro vibrations. As a result, these frameworks are not directly applicable to our synthesis task, where high-fidelity signal generation that preserves micro-scale phase variations is essential.

**Diffusion models and rectified flow.** Diffusion generative models (Ho et al., 2020; Sohl-Dickstein et al., 2015) generate data by learning to reverse a predefined noising process, starting from Gaussian noise, and iteratively denoising toward the data distribution. This paradigm has achieved significant success in image/video synthesis (Bao et al., 2023; Blattmann et al., 2023) and time-series generation (Chen et al., 2025; Zhang et al., 2025a). Recently, rectified flow models (Albergo & Vanden-Eijnden, 2023; Lipman et al., 2023; Liu et al., 2022) have emerged to refine the generative process by establishing a transport map between two probability distributions. By learning a velocity field, these models are often easier to train and sample than traditional probability flow ODEs, and have started to gain attention in the time-series domain (Wei et al., 2026).

Furthermore, flow matching is not restricted to Gaussian priors and can transport between arbitrary source and target distributions in principle. Recent extensions along this line include image-to-image translation via conditional/paired transport (Liu et al., 2023) and Schrödinger-bridge-based formulations (Liu et al., 2024; De Bortoli et al., 2023). Besides in mapping across different data distributions (Liu et al., 2025), we adopts this trajectory to enable synthesis by directly transforming ECG latent representations into the radar vibration domain.

## 3. Methodology

This section presents the proposed **Cardio-mmFlow** framework and its core components, with the overall architecture shown in Figure 1. Sec. 3.1 introduces the direct latent flow matching model for ECG-to-radar latent transport. Sec. 3.2 describes the physics-inspired MSD-FiLM module for metadata-conditioned modulation. Sec. 3.3 presents the optimization objectives, including the Wasserstein distance constraint, frequency-domain loss, and Classifier-Free Guidance (CFG).

### 3.1. VAE and Cross-domain Flow Matching

Although both ECG and radar are time-series, they belong to fundamentally different sensing modalities: ECG records cardiac electric potentials, whereas mmWave radar captures vibrations induced phase modulations of the thorax. To bridge this modality gap, we depart from the conventional noise-to-signal paradigm and instead propose a *Gaussian-prior-free direct flow matching* between their latent manifolds. Crucially, since ECG and radar segments are synchronized recordings of the cardiac cycle, the ECG latent offers an informative and physiology-grounded initialization for direct latent evolution. On the other hand, an unstructured Gaussian noise prior is forcing the transport to learn modality alignment from a non-informative starting point.

To ensure robust trajectory learning, we represent the source as a distribution rather than a single deterministic code. Given an input ECG segment $x$, the ECG VAE encoder $E_{ecg}$ predicts $(\boldsymbol{\mu}_{ecg}, \boldsymbol{\sigma}_{ecg}^2)$ and induces $q_{ecg}(\mathbf{z} \mid x) = \mathcal{N}(\boldsymbol{\mu}_{ecg}, \mathrm{diag}(\boldsymbol{\sigma}_{ecg}^2))$. We sample $\mathbf{z}_{ecg} \sim q_{ecg}(\mathbf{z} \mid x)$ to obtain a distribution of plausible starting states, which helps regularizes the transport. For the target endpoint, the radar VAE encoder $E_r$ maps the paired radar segment $y_r$ to a radar latent $\mathbf{z}_r \sim q_r(\mathbf{z} \mid y_r) \in \mathcal{Z}_r$. Rather than modifying the pretrained ECG encoder, we append a learnable projector $P_\phi$ to align $\mathbf{z}_{ecg}$ toward the radar-latent coordinate system as $\mathbf{z}'_{ecg} \triangleq P_\phi(\mathbf{z}_{ecg})$. We jointly optimize $P_\phi$ and the flow model so that $P_\phi$ handles distribution-level alignment, while the flow captures the remaining conditional transport (Sec. 3.3).

**Velocity Field and Radar Decoding.** We model the cross-modal mapping as a continuous-time latent flow, from the distribution-aligned source $\mathbf{z}'_{ecg}$ at $t = 0$ to the target radar

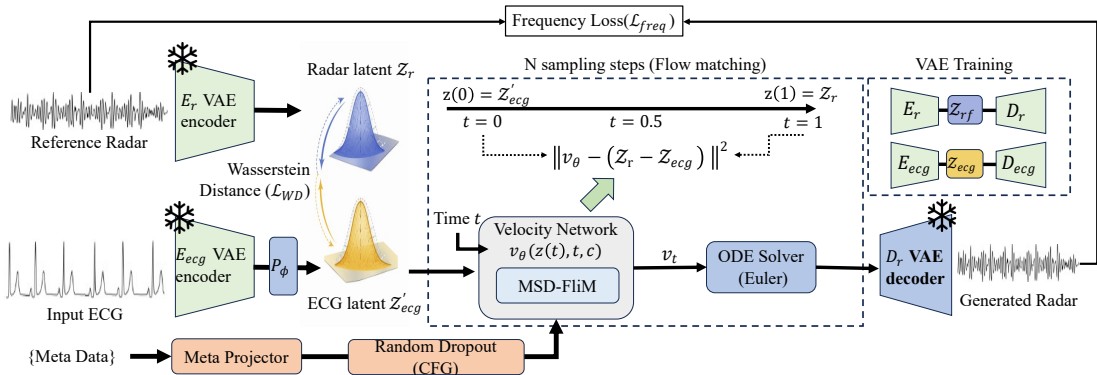

**Figure 1.** **Framework design of Cardio-mmFlow.** Frozen VAEs define modality-specific latent manifolds. A latent projector aligns two latents using $\mathcal{L}_{\mathrm{WD}}$, then a Transformer velocity network with MSD-FiLM learns the flow-matching transport from $\mathbf{z}(0)$ to $\mathbf{z}(1)$, conditioned on subject metadata via meta-projectors and CFG. $\mathcal{L}_{\{\nabla\}\amalg}$ further constrain the velocity network in spectrum.

latent $\mathbf{z}_r \in \mathcal{Z}_r$ at $t = 1$.

Given the endpoint latents, we consider a linear latent path:

$$\mathbf{z}_t = (1-t)\mathbf{z}'_{ecg} + t\,\mathbf{z}_r, \qquad t \in [0,1], \qquad (2)$$

Then the corresponding vector field with a Transformer backbone by defining the ODE can be parameterized as

$$\frac{d\mathbf{z}(t)}{dt} = v_\theta\big(\mathbf{z}(t), t, \mathbf{c}\big), \qquad \mathbf{z}(0) = \mathbf{z}'_{ecg}, \qquad (3)$$

where $\mathbf{c}$ is the bio-condition embedding instantiated by the MSD-FiLM modulation and the Meta Projector (Sec. 3.2). After integrating the trajectory to $t = 1$, we obtain the generated terminal latent $\hat{\mathbf{z}}_r \triangleq \mathbf{z}(1)$, which is decoded back to the signal space using the pre-trained radar decoder:

$$\hat{\mathbf{y}}_r = D_r\big(\hat{\mathbf{z}}_r\big). \qquad (4)$$

This decoding step reconstructs the synthesized radar waveform from the evolved latent state in $\mathcal{Z}_r$.

### 3.2. MSD-FiLM Module

We model the propagation of cardiac-induced microvibrations from the heart to the thoracic surface using a **MSD system**. Specifically, the chest-wall displacement $d(t)$ driven by the internal cardiac excitation $F(t)$ can be described by the following second-order differential equation:

$$m\ddot{d}(t) + c\dot{d}(t) + kd(t) = F(t), \qquad (5)$$

where $m$, $c$, and $k$ denote the effective mass, damping coefficient, and stiffness of thoracic tissues respectively.

However, directly identifying the full MSD parameters $(m, c, k)$ is impractical in our setting. Although this is a driven system ($F(t) \neq 0$), we did a transient-envelope analysis (see details in *Appendix* A.1) and found that the driven response can be viewed through the impulse-response

ring-down, whose envelope decay is governed by the *effective damping rate*. We therefore adopt a **damping-centric** simplification and summarize propagation variability with an attenuation scale that can be stably injected as a low-dimensional prior during flow evolution.

Consistent with this view, we parameterize a modulation gain for dissipative attenuation. Let $\tau$ be the flow time and $\gamma$ a subject-specific conditioning feature capturing propagation variability. The raw projection are defined as

$$\mathbf{s} = \mathbf{W}_s[\tau; \gamma] + \mathbf{b}_s, \qquad (6)$$

where $[\tau; \gamma]$ denotes concatenation. We then compute a non-negative damping intensity via the *softplus* function,

$$damp = \ln\big(1 + \exp(\mathbf{s})\big), \quad \boldsymbol{\alpha} = \exp(-damp), \qquad (7)$$

which guarantees $damp \geq \mathbf{0}$ and bounds the element-wise attenuation gain as $\boldsymbol{\alpha} \in (0,1]^D$, matching the monotone energy dissipation behavior in biological tissues. We obtain $\gamma$ from subject metadata using a lightweight Meta Projector.

The Meta Projector maps metadata into a personalized embedding $\gamma \in \mathbb{R}^D$. Raw metadata $[g, a, h, w, b]$ (gender, age, height, weight, BMI) are mapped into the conditional embedding, where $\mathbb{R}^D$ denotes the $D$-dimensional real-valued vector space. For continuous variables $x \in \{a, h, w, b\}$, we first apply physical boundaries $[x_{\min}, x_{\max}]$ and normalize them to $[0, 1]$ via

$$\bar{x} = \frac{\mathrm{clip}(x; x_{\min}, x_{\max}) - x_{\min}}{x_{\max} - x_{\min}}, \qquad (8)$$

Each normalized scalar is encoded using low-dimensional Gaussian Fourier features $\phi(\bar{x}) \in \mathbb{R}^8$, while the discrete gender $g$ is mapped to an embedding $\mathbf{e}_g \in \mathbb{R}^8$. Concatenating all components yields $\mathbf{z}_{in} \in \mathbb{R}^{40}$, which is processed by a narrow-bottleneck MLP to produce the final embedding:

$$\gamma = \mathrm{LN}\Big(\mathbf{W}_2\,\mathrm{SiLU}\big(\mathrm{Dropout}(\mathbf{W}_1\mathbf{z}_{in})\big)\Big), \qquad (9)$$

where $\mathbf{W}_1, \mathbf{W}_2$ are learnable linear projections, $\mathrm{Dropout}(\cdot)$ denotes standard dropout, $\mathrm{SiLU}(\cdot)$ is the Sigmoid Linear Unit activation, and $\mathrm{LN}(\cdot)$ is LayerNorm. The conditional embedding $\boldsymbol{\gamma}$ is injected into the Transformer backbone via the FiLM module. Given an intermediate feature tensor $\mathbf{H}$, FiLM applies an attenuation–shift modulation:

$$\mathrm{FiLM}(\mathbf{H}; \tau, \boldsymbol{\gamma}) = \mathrm{LN}(\mathbf{H}) \odot \boldsymbol{\alpha} + \boldsymbol{\beta}(\tau), \qquad (10)$$

where $\tau$ is the flow-time embedding, $\mathrm{LN}(\cdot)$ denotes Layer-Norm over the feature dimension, and $\boldsymbol{\alpha}$ (from Eq. 7) and $\boldsymbol{\beta}$ are broadcast to match the shape of $\mathbf{H}$. The shift term $\boldsymbol{\beta}(\tau)$ is decoupled from subject metadata and depends only on $\tau$:

$$\boldsymbol{\beta}(\tau) = \mathrm{clip}\left((\mathbf{W}_b \tau + \mathbf{b}_b), -1, 1\right). \qquad (11)$$

This step prevents subject-specific offsets from being encoded in $\boldsymbol{\beta}$, thereby forcing individual variability to be explained primarily via the damping-driven attenuation in $\boldsymbol{\alpha}$.

### 3.3. Training the Cardio-mmFlow

The training is conducted in two distinct stages: (i) pre-training the modality-specific VAEs to establish structured manifolds, and (ii) optimizing the flow velocity field with a multi-objective strategy.

**Stage I: VAE Pre-training.** We independently pre-train modality-specific VAEs for ECG and radar to map heterogeneous signals into a latent space $\mathcal{Z}$. For each modality, we optimize the standard evidence lower bound:

$$\mathcal{L}_{\mathrm{VAE}} = \mathbb{E}_{z \sim q(z|x)}\left[\|x - \hat{x}\|_2^2\right] + \lambda_{\mathrm{KL}} D_{\mathrm{KL}}\left(q(z \mid x) \| p(z)\right), \qquad (12)$$

where $z \in \mathcal{Z}$ is sampled from $q(z \mid x)$ and $\hat{x} = D(z)$. The first term is the reconstruction loss, and the second term regularizes the latent distribution. After pre-training, we freeze encoders for the subsequent flow optimization stage.

**Stage II: Training Flow Matching.** We train a conditional velocity field $v_\theta$ to transport marginally aligned ECG latents toward radar latents (conditioned on metadata). We jointly optimize a latent aligner $P_\phi$ and a conditional velocity field $v_\theta$ via the flow matching loss ($\mathcal{L}_{\mathrm{FM}}$):

$$\mathcal{L}_{\mathrm{FM}} = \mathbb{E}_{((x_0,x_1), t)}\left[\left\|v_\theta(z_t, t, c) - (z_1 - \tilde{z}_0)\right\|_2^2\right], \quad (13)$$

where $z_0 \sim q_{\mathrm{ecg}}(z \mid x_0)$ and $\tilde{z}_0 = P_\phi(z_0)$, and $z_1 = E_r(x_1)$ is kept deterministic as an anchor. The $t$ is sampled from $\mathcal{U}(0, 1)$ and form the linear interpolation $z_t = (1 - t)\tilde{z}_0 + t z_1$, where $c$ denotes the conditioning signal.

**Distribution Pre-Alignment ($\mathcal{L}_{\mathrm{WD}}$).** A key role of the latent projector $P_\phi$ is to perform marginal distribution alignment by project ECG latents into the radar latent space, thereby reducing the mismatch induced by independently pretrained VAEs. Accordingly, we impose a Wasserstein

alignment (WD) (Arjovsky et al., 2017) loss between the mini-batch empirical distributions of the projected source latent $\tilde{z}_0$ and the target radar latent $z_1$:

$$\mathcal{L}_{\mathrm{WD}} = W_1\left(\widehat{\mathbb{P}}_{\tilde{z}_0}, \widehat{\mathbb{P}}_{z_1}\right), \qquad (14)$$

where $\widehat{\mathbb{P}}_{\tilde{z}_0}$ and $\widehat{\mathbb{P}}_{z_1}$ denote the corresponding mini-batch empirical distributions. Importantly, $\mathcal{L}_{\mathrm{WD}}$ performs *distribution-level* marginally matching rather than enforcing point-wise correspondence between paired samples. Otherwise, $P_\phi$ could degenerate into a trivial per-sample mapping that directly reconstructs radar latents, undermining the subsequent transport learning. In practice, $W_1$ is efficiently approximated via Sinkhorn iterations (see *Appendix A.3*). This pre-alignment simplifies velocity-field learning by separating coarse marginal alignment (handled by $P_\phi$) from transport dynamics (modeled by the flow).

**Spectral Fidelity Loss ($\mathcal{L}_{\mathrm{freq}}$).** Let $\hat{\mathbf{x}}_1 = D_{\mathrm{rf}}(\hat{\mathbf{z}}_1)$, and $\mathcal{F}$ denote the real FFT along the temporal dimension. The $\hat{\mathbf{z}}_1$ is the generated radar latent obtained by solving the flow ODE under $v_\theta$ from $\tilde{\mathbf{z}}_0$ to $t = 1$. To preserve relative spectral energy, we normalize both magnitudes using a shared factor $s \triangleq \max(|\mathcal{F}(\mathbf{x}_1)|) + \varepsilon$ ($\varepsilon$ is a small constant to avoid division by zero). We then define the normalized magnitude spectrum $S(\cdot)$ and the phase spectrum $\Phi(\cdot)$ as

$$\begin{cases} S(\mathbf{x}) \triangleq \dfrac{|\mathcal{F}(\mathbf{x})|}{s}, \\ \Phi(\mathbf{x}) \triangleq \angle \mathcal{F}(\mathbf{x}). \end{cases} \qquad (15)$$

The spectral loss is then defined as,

$$\begin{aligned} \mathcal{L}_{\mathrm{freq}} = \ &\|S(\hat{\mathbf{x}}_1) - S(\mathbf{x}_1)\|_2^2 \\ &+ \lambda_p \Big(1 - \cos\big(\mathrm{vec}(\Phi(\hat{\mathbf{x}}_1)), \mathrm{vec}(\Phi(\mathbf{x}_1))\big)\Big). \end{aligned} \qquad (16)$$

where $\cos(\mathbf{a}, \mathbf{b})$ denotes cosine similarity, and $\lambda_p$ lightly weights the phase term (we use $\lambda_p = 0.5$).

Finally, the Stage-II objective is:

$$\mathcal{L}_{\mathrm{total}} = \mathcal{L}_{\mathrm{FM}} + \lambda_1 \mathcal{L}_{\mathrm{WD}} + \lambda_2 \mathcal{L}_{\mathrm{freq}}. \qquad (17)$$

**Classifier-Free Guidance (CFG).** To further improve fidelity and adherence to subject-specific biomechanical traits, we adopt CFG method. During Stage-II training, the meta embedding $c$ is randomly replaced by a null token $\varnothing$ with a fixed probability, enabling the model to learn both conditional and unconditional velocity fields. At inference, the velocity are computed as

$$\hat{v}_\theta(z_t, t) = v_\theta(z_t, t, \varnothing) + \omega\Big(v_\theta(z_t, t, \mathbf{c}) - v_\theta(z_t, t, \varnothing)\Big), \qquad (18)$$

where $\omega \geq 1$ is the guidance scale. Intuitively, a larger $\omega$ strengthens condition adherence, and encourages the generated trajectory to better conform to the personalized damping system specified by the MSD-FiLM.

# 4. Experiments

In this section, we evaluate Cardio-mmFlow from three complementary perspectives: synthesis fidelity, downstream utility, and model behavior. Specifically, we first report the experimental setup and implementation details, followed by baseline comparisons and ablation studies. We further assess downstream utility through zero-shot mmWave→ECG reconstruction, with additional studies on Atrial Fibrillation (AFib) classification, latent-space interpolation, and metadata-driven modulation.

## 4.1. Implementation Details

**Experimental Setup.** Since ECG to mmWave data synthesis remains largely unexplored and lacks task-specific prior work, direct within-domain baseline comparisons are currently unavailable. To fairly evaluate our framework, we adapt and repurpose two categories of generative models: *(i)* foundational methods like (cVAE(Sohn et al., 2015), cGAN(Isola et al., 2017)), standard Diffusion transformer(Peebles & Xie, 2023a), and *(ii)* recent state-of-the-art bio-signal transformation models (RDDM (Shome et al., 2024) and PPGflowECG (Fang et al., 2025)) originally designed for related tasks (e.g., PPG→ECG). For a rigorous comparison, all baselines are trained on the identical datasets and pre-processing pipelines. All methods are adapted to conditioned on ECG and metadata, and trained to generate **mmWave radar** signals (details in *Appendix* A.4).

**Model Architecture.** The Radar/ECG VAEs are constructed using ResNet blocks that map signals into a 16-dimensional latent space. The flow-matching backbone is a 4-layer Diffusion Transformer (DiT) (Peebles & Xie, 2023b) with a hidden dimension of 256 and 8 attention heads. Each block within the backbone utilizes Transformer (Vaswani et al., 2017) layers using fixed sinusoidal positional embeddings. The Latent Projector consists of two convolution layers followed by normalization. The Meta Projector is a width-40 MLP that maps metadata to a conditioning vector, which is then fed into the MSD-FiLM module for modulation. Time steps are embedded with adaLN-Zero. An Euler solver is used for flow ODE. Details in *Appendix* A.7.

**Dataset.** We use four datasets (3 Public/ 1 self). (i) A dataset of clinically recorded radar vital signs with synchronized reference sensor (include ECG) signals (Schellenberger et al., 2020), containing 30 subjects (14 male/16 female) with ~24-hour recordings under five scenarios. (ii) PTB-XL (Wagner et al., 2020), a large-scale clinical ECG dataset with 21,799 10s records from 18,869 patients. (iii) A sub dataset (20 subjects) of synchronized collected radar and ECG with Atrial fibrillation (Af)(Yuan et al., 2025). (iv) A self-collected synchronized ECG and mmWave radar dataset, including 10 subjects (6 male/4 female) with ~3-hour recordings in sleeping condition (details in *Appendix*.A.2).

Dataset (i) is used for model training (Stage I/II), dataset (ii) provides diverse ECG inputs for large-scale synthesis, and (iii) is used for downstream zero-shot classification task. Dataset (iv) is used exclusively for downstream evaluation. All dataset are resampled to 200Hz and normalize by subjects z-score. Then a band-pass filter (10-40Hz) with 4th order Butterworth process the radar signal. The ECG are cleaned using NeuroKit Lib (Makowski et al., 2021). A sliding window cut both paired signal length to 512 points. Both stages use subject-disjoint split to prevent identity leakage. Users used in training will be removed in validation.

**Training Details.** Both training stages use batch size 512. We use AdamW ($\beta_1 = 0.9$, $\beta_2 = 0.999$, weight decay $= 1 \times 10^{-4}$) with a cosine learning-rate schedule. Each modality-specific VAE is trained for 300 epochs with a 20-epoch warmup ($lr = 1 \times 10^{-4}$) and $\lambda_{\mathrm{KL}} = 0.1$. In optimizing the flow model, model are trained with 300 epochs and $lr = 5 \times 10^{-4}$ and a 9,100 step warmup period. The weights for $\mathcal{L}_{\mathrm{WD}}$ and $\mathcal{L}_{\mathrm{freq}}$ are set to 0.1 and 0.5 respectively. The condition dropout rate of 0.3 is applied to realize CFG. A 10 steps ODE solver is used. All hyper-params are set from serval manually test attempts.

**Evaluation Metrics.** For synthesis quality evaluation, we report four complementary metrics that probe distributional realism, spectral consistency, waveform fidelity, and physiological readability: (i) Latent-FID measures the distance between real and synthesized latent feature distributions using pooled latents from a frozen radar VAE encoder, serving as a feature-level proxy for realism and diversity; (ii) PSD similarity compares the normalized power spectral densities of real and synthesized signals to verify whether dominant frequency components and relative spectral shapes are preserved; (iii) time-domain MSE quantifies sample-wise waveform discrepancy; and (iv) HR-AE measures the absolute heart-rate estimation error to evaluate whether the synthesized signals retain physiologically interpretable periodicity.

For downstream mmWave→ECG reconstruction, we report RMSE, Pearson correlation coefficient (PCC), and cardiac event timing error. The timing error is defined as the mean absolute error of the detected R–R intervals between the predicted and reference ECGs, reflecting whether the reconstructed ECG preserves beat-level temporal structure. Details are provided in Appendix A.5.

## 4.2. Overall Results of Synthesis with Baselines.

**Table 1** summarizes the baseline comparison on radar synthesis quality. Ours achieves the best synthesis fidelity across the three primary quality metrics, obtaining the lowest Latent-FID and MSE, as well as the highest PSD similarity. This indicates improved distributional realism time-domain reconstruction accuracy and spectral consistency. In terms of physiological utility, our model yields a com-

*Table 1.* Baseline comparison on radar synthesis quality. ↓ lower is better; ↑ higher is better. **Bolds** are the best while underline is the second best results.

| Method | Latent-FID ↓ | MSE ↓ | PSD Sim. ↑ | HR-AE (bpm) ↓ | #Params (M) |
|---|---|---|---|---|---|
| cGAN | 15.21 | $0.036 \pm 0.019$ | $0.871 \pm 0.085$ | 12.56 | 10.7M |
| cVAE | 29.31 | $0.032 \pm 0.015$ | $0.823 \pm 0.110$ | 16.72 | 14.4M |
| DiT | 16.79 | $0.051 \pm 0.021$ | $0.905 \pm 0.069$ | 12.21 | 13.7M |
| RDDM | 23.20 | $0.065 \pm 0.196$ | $0.850 \pm 0.128$ | 11.27 | 12.4M |
| PPGflowECG | 18.72 | $0.029 \pm 0.014$ | $0.864 \pm 0.107$ | **6.00** | 11.8M |
| **Ours** | **14.98** | $\mathbf{0.027 \pm 0.013}$ | $\mathbf{0.919 \pm 0.078}$ | 6.66 | 11.2M |

*Table 2.* mmWave→ECG results under the **Zero-shot** protocol. **Bolds** are the best while underline is the second best results.

| Method | RMSE ↓ | PCC ↑ | RR Err. (ms) ↓ |
|---|---|---|---|
| cGAN | $0.13 \pm 0.03$ | $0.71 \pm 0.07$ | $11.92 \pm 3.53$ |
| cVAE | $0.17 \pm 0.03$ | $0.51 \pm 0.15$ | $15.37 \pm 7.45$ |
| DiT | $0.12 \pm 0.03$ | $0.74 \pm 0.08$ | $9.17 \pm 4.98$ |
| RDDM | $0.19 \pm 0.04$ | $0.57 \pm 0.15$ | $16.01 \pm 6.30$ |
| PPGflowECG | $0.13 \pm 0.02$ | $0.69 \pm 0.06$ | $9.28 \pm 4.67$ |
| Our | $\mathbf{0.11 \pm 0.02}$ | $\mathbf{0.79 \pm 0.06}$ | $\mathbf{8.13 \pm 3.94}$ |

*Table 3.* Calibration ratio ablation (AirECG). All results are displayed as relative changes

| Metric | 10% | 20% | 30% | 40% |
|---|---|---|---|---|
| RMSE | $-0.49\%$ | $-0.59\%$ | $-2.40\%$ | $-1.03\%$ |
| PCC | $+4.17\%$ | $+1.80\%$ | $+1.37\%$ | $+2.95\%$ |
| RR Err. (ms) | $+1.71\%$ | $+0.52\%$ | $-2.18\%$ | $-3.64\%$ |

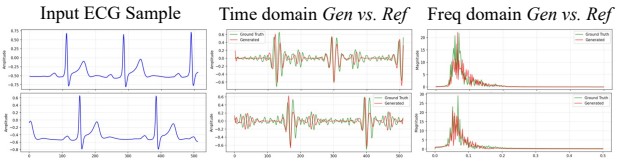

*Figure 2.* Samples of Cardio-mmFlow synthetic data and ground truth with time/frequency domain comparison.

petitive HR-AE, slightly higher than PPGflowECG but substantially better than the others. Notably, these gains are achieved with a comparable model size, suggesting that the improvements are not solely attributable to larger capacity. Figure 2 presents representative synthesized samples. The results show spectral profiles consistent with the ground truth, while maintaining comparable period of cardiac cycle and signal morphology.

### 4.3. Downstream Tasks Evaluation.

To evaluate the practical value of synthesized data beyond perceptual similarity, we benchmark a representative downstream task: **mmWave → ECG reconstruction**. This task is limited by the high cost of collecting large-scale mmWave–ECG pairs and by cross-subject generalization, making it a suitable testbed for synthetic supervision.

**Dataset and Training Protocol.** We use our self-collected synchronized mmWave–ECG dataset for downstream evaluation and adopt a subject-disjoint split to prevent identity leakage and assess cross-subject transfer. For each trial, we randomly select one subject as the target user for testing. Personalized synthetic supervision is generated by conditioning on the target's metadata, with **ECG templates** from PTB-XL (NORM) as the physiological basis.

We consider two protocols: (i) a *zero-shot* setting, where the downstream model is trained only on synthetic pairs and evaluated on real target-user data without using any real target-user samples; and (ii) a *few-shot calibration* setting, where the same model is fine-tuned with 10–40% paired real data from the target user and tested on the held-out remainder. The zero-shot protocol is our primary evaluation because it directly tests synthetic-to-real transfer, while the few-shot protocol provides a complementary view of the model behavior when limited user-specific calibration is available. We use AirECG (Zhao et al., 2024a) as the

downstream model, a high-capacity DiT-based diffusion framework, and keep the split subject-disjoint to isolate the effect of synthetic augmentation.

**Evaluation results.** We evaluate all baseline synthetic data on the downstream radar-to-ECG task and report RMSE, PCC, and R–R peak timing error. As shown in Table 2, our method achieves the best overall zero-shot performance, securing the lowest RMSE and RR error, while attaining the highest PCC. Among the baselines, DiT and PPGflowECG provide competitive reconstruction quality. In contrast, RDDM performs notably worse across all metrics. Although cGAN achieves moderate waveform correlation, its limited temporal precision leads to a substantially higher RR error.

Table 3 reports relative changes over the zero-shot setting under different calibration ratios. Few-shot calibration provides incremental refinement on top of the zero-shot model. PCC improves across all ratios, and RMSE is consistently reduced, showing that limited real target-user data can further improve waveform reconstruction. RR error exhibits small fluctuations at low calibration ratios, but improves when 30%–40% calibration data is used. Overall, these results suggest that user-specific calibration further refines reconstruction performance, while the relatively modest gains indicate that the synthesized data have substantially reduced the cross-subject transfer gap before calibration. We note that the calibration gain may also be affected by the downstream model design and fine-tuning strategy.

### 4.4. Ablation Study

All the results are inference on the validation dataset (user not appear in training) with 10 steps ODE solver.

**Latent init vs. noise init.** We replace the latent-prior start with Gaussian noise ($z_0 \sim \mathcal{N}(0, I)$) to simulate a standard diffusion setup. Table 4 reveals that ECG-latent initial-

*Table 4.* Ablation studies on different components. Vertical lines separate distinct ablation experiments. Evaluated on 5,632 paired segments. ↓ lower is better; ↑ higher is better. Model best results are marked in **bold**.

| Metric | Initialization Strategy | | Modulation Layer | | | Loss Components | | Guidance Strategy | |
|---|---|---|---|---|---|---|---|---|---|
| | Latent | Noise | MSD-FiLM | MLP-FiLM | AdaLN + crossAttn | w/o $\mathcal{L}_{\text{freq}}$ | w/o $\mathcal{L}_{\text{WD}}$ | w/ CFG | w/o CFG |
| MSE ↓ | **0.027**±**0.013** | 0.031±0.014 | 0.027±0.013 | 0.032±0.014 | **0.026**±**0.014** | 0.029±0.014 | 0.035±0.015 | **0.027**±**0.013** | 0.029±0.015 |
| HR-AE ↓ | **6.66** | 12.95 | **6.66** | 11.39 | 7.58 | 8.75 | 9.96 | **6.66** | 8.82 |
| PSD Sim ↑ | **0.919**±**0.078** | 0.807±0.100 | **0.919**±**0.078** | 0.825±0.112 | 0.819±0.082 | 0.879±0.098 | 0.846±0.103 | **0.919**±**0.078** | 0.919±0.068 |
| Latent-FID ↓ | **14.98** | 24.55 | **14.98** | 18.273 | 24.93 | 18.09 | 24.64 | **14.98** | 18.11 |

ization offers a clear advantage, decreasing Latent-FID by 38.2% and boosting PSD similarity by 13.9%. This suggests that initiating from a physiological manifold acts as a robust anchor, ensuring synthesized signals align more closely with real radar statistics than those evolved from noise.

**Impact of Modulation.** To validate the MSD-inspired modulation design, we compare MSD-FiLM with two generic metadata-injection alternatives while keeping other components unchanged. Vanilla MLP-FiLM directly predicts FiLM parameters from metadata without the attenuation constraint, while AdaLN + crossAttn uses generic attention-based metadata conditioning. As shown in Table 4, both generic alternatives degrade distributional fidelity. Compared with MSD-FiLM, Vanilla MLP-FiLM increases Latent-FID from 14.98 to 18.273, while AdaLN + crossAttn further increases it to 24.93. For HR-AE and PSD similarity, MSD-FiLM consistently outperforms both alternatives, although AdaLN + crossAttn obtains a slightly lower MSE. These results suggest that the structured attenuation prior in MSD-FiLM provides more effective metadata modulation than generic conditioning mechanisms.

**Impact of $\mathcal{L}_{\text{WD}}$ and $\mathcal{L}_{\text{freq}}$.** We ablate the two objectives by removing $\mathcal{L}_{\text{freq}}$ or $\mathcal{L}_{\text{WD}}$ (Table 4). Dropping $\mathcal{L}_{\text{freq}}$ affects Latent-FID to a lesser extent (18.09), but it noticeably degrades time-domain error (higher MSE) and spectral similarity (lower PSD similarity), indicating that $\mathcal{L}_{\text{freq}}$ mainly serves as a fine-grained temporal–spectral regularizer. In contrast, removing $\mathcal{L}_{\text{WD}}$ causes a much larger distribution mismatch, increasing Latent-FID to 24.64 and producing less stable synthesis (missing frequency components see *Appendix* B). This suggests that $\mathcal{L}_{\text{WD}}$ anchors latent-space alignment, while $\mathcal{L}_{\text{freq}}$ primarily refines spectral details.

**Impact of CFG.** We ablate classifier-free guidance by removing masking, while keeping the training setup and conditioning inputs unchanged. As shown in Table 4, removing CFG leads to worse latent distribution alignment, indicating reduced adherence to the desired conditional distribution. Meanwhile, MSE/PSD remain comparable which is expected since point-wise metrics are less discriminative for radar waveforms that contain many subtle noise-like fluctuations. Overall, CFG mainly benefits conditional controllability and global distribution matching.

### 4.5. Additional Evaluation Results

**Utility in AFib Classification.** To further examine the cardiac abnormality detection utility of the synthesized radar signals under scarce pathological recordings, we evaluate a downstream AFib detection task with a strict zero-shot protocol. Specifically, we synthesize an AFib radar dataset from PTB-XL AFib ECG samples and train a ResNet-18 classifier solely on the synthetic radar data. The trained classifier is then evaluated on a separate real radar dataset without using any real AFib radar samples for training. Implementation details are provided in *Appendix* A.6.

*Table 5.* Downstream AFib classification performance. Results are reported as mean ± std over five random seeds. ↑ indicates higher is better.

| Metric ↑ | Mean ± Std |
|---|---|
| Recall | 0.87 ± 0.039 |
| Precision | 0.62 ± 0.046 |
| F1-Score | 0.72 ± 0.038 |
| Accuracy | 0.68 ± 0.040 |
| AUC-ROC | 0.72 ± 0.043 |

We repeat the evaluation with five random seeds and report mean ± std in Table 5. The model achieves a high recall of 87.3%, indicating that the synthetic radar data preserves AFib-discriminative patterns that can transfer to real recordings. Precision is relatively lower at 62.1%, which is expected under the synthetic-to-real domain gap and suggests that the classifier tends to produce more positive predictions. Nevertheless, the F1 score and AUC-ROC show that the learned representations retain useful discriminative capability for zero-shot AFib classification. This high-recall/moderate-precision behavior is meaningful for monitoring scenarios, where sensitivity is often prioritized to reduce missed intermittent AFib events. The false positives can be further handled by follow-up measurements or clinical confirmation.

**VAE reconstruction quality.** We evaluate the modality-specific VAEs to verify the fidelity of the latent representations used in the proposed ECG-to-radar synthesis framework. Since the cross-modal generator operates in the latent space, the ECG VAE is expected to preserve cardiac morphology before translation, while the Radar VAE determines

*Table 6.* Reconstruction quality evaluation of modality-specific VAEs. RMSE measures point-wise reconstruction error, while PCC measures waveform correlation.

|  | RMSE ↓ | PCC ↑ |
|---|---|---|
| ECG VAE | $0.024 \pm 0.008$ | $0.995 \pm 0.004$ |
| Radar VAE | $0.039 \pm 0.023$ | $0.954 \pm 0.041$ |

whether generated latent codes can be decoded into realistic radar waveforms. As shown in Table 6, both VAEs achieve low reconstruction errors and high waveform correlations, demonstrating that the learned latent spaces provide a faithful basis for subsequent cross-modal generation.

**Latent-space interpolation.** We linearly interpolate between two latent codes corresponding to heart rates from 58.5 to 90.1 bpm under a fixed subject metadata profile: male, age 40, height 184 cm, weight 74 kg, and BMI 21.85. As shown in Figure 8, each panel presents the synthesized waveform at an interpolation coefficient $\alpha \in [0, 1]$, where $\alpha = 0$ and $\alpha = 1$ denote the two endpoints. The waveforms change smoothly along the interpolation trajectory, illustrating continuous and controllable modulation of radar rhythm characteristics. Full traces and additional details are provided in *Appendix* B.

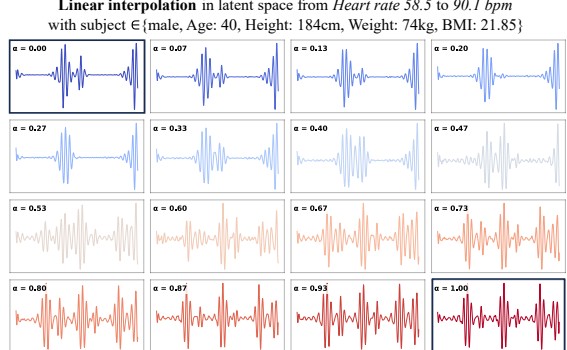

*Figure 3.* Linear interpolation between different Heart rate.

**Metadata modulation.** We vary individual attributes to test the metadata modulation effect. Specifically, we fix other metadata and adjust one attribute within a reasonable range to observe how the model output changes. As shown in Figure 4, *Height* and *Weight* mainly influence amplitude attenuation, while *BMI* steers fine-grained morphology.

## 5. Conclusion

In this work, we introduced **Cardio-mmFlow**, a physics-informed generative framework which synthesize mmWave radar from clinical ECG templates. It aims to overcome the critical data scarcity and generalization bottlenecks in contactless ECG sensing. By integrating Gaussian prior free latent flow matching with physical inspired MSD-FiLM,

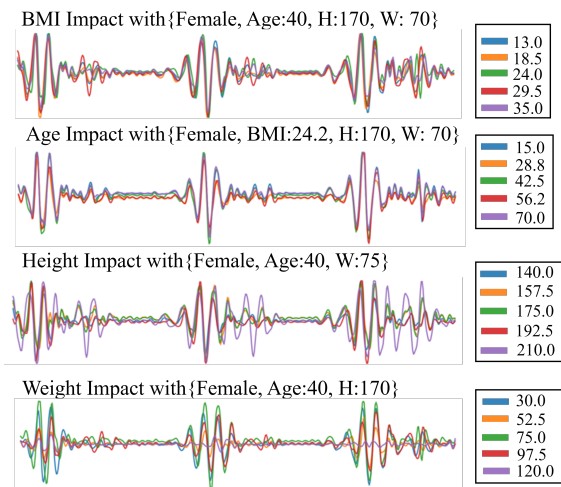

*Figure 4.* Metadata elements modulation test.

we establish a direct cross-modal mapping that captures subject-specific signal propagation. Our approach offers a scalable solution to the data bottleneck, toward more reliable deployment of non-contact ECG monitoring systems.

## Acknowledgments

The authors sincerely thank the anonymous reviewers for their constructive suggestions, which helped improve the presentation of this work. This research was supported in part by the National Natural Science Foundation of China under Grant 62472366; the Project of Department of Education of Guangdong Province under Grants 2023KCXTD042, 2024GCZX003, 2024KCXTD008, and 2025KCXTD056; the Guangdong Provincial Key Laboratory of Integrated Communication, Sensing and Computation for Ubiquitous Internet of Things under Grant 2023B1212010007; the 111 Center under Grant D25008; and the Shenzhen Science and Technology Foundation under Grant ZDSYS20190902092853047.

## Impact Statement

This paper presents work whose goal it to advance contactless cardiac sensing by synthesizing mmWave radar signals from widely available ECG data. The proposed framework may help alleviate the need for large-scale synchronized radar–ECG collection and support research on non-contact physiological monitoring in home and healthcare scenarios. For practical deployment, a small amount of user-specific real data is still recommended for calibration under target sensing conditions. Accordingly, the synthesized signals serve as a scalable complementary resource for training and benchmarking, reducing reliance on extensive real recordings without replacing or overlooking the role of real data.

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

# A. APPENDIX

## A.1. Why the Full MSD System Can Be Simplified for Attenuation Modeling

This appendix supports Sec. 3.2 by clarifying why beat-wise cardiac micro-vibrations can be treated as a driven yet transient process for the purpose of modeling attenuation.

**Driven-Transient Dynamics.** Cardiac-induced surface micro-vibrations are non-stationary and beat-wise. The effective cardiac excitation force $F(t)$ is physically concentrated in short mechanical events such as valve closures. The full driven system is governed by:

$$m\ddot{d}(t) + c\dot{d}(t) + kd(t) = F(t). \tag{19}$$

To analyze the signal envelope, consider a single heartbeat where the excitation is localized. There exists an interval $[t_0, t_1]$ such that $F(t) \neq 0$ (energy injection), followed by a period $t > t_1$ where $F(t) \approx 0$ (post-excitation ring-down). While the oscillatory content is shaped by the stiffness–mass relation, excitation pattern, and other waveform factors that are implicitly learned by the neural backbone, amplitude attenuation is most explicitly reflected in the post-excitation ring-down phase.

**Envelope decay of the homogeneous response.** For the post-excitation ring-down interval ($t > t_1$), the excitation becomes negligible ($F(t) \approx 0$). Eq. (19) can then be approximated by the homogeneous ODE:

$$m\ddot{d}(t) + c\dot{d}(t) + kd(t) = 0. \tag{20}$$

Let $\tau = t - t_1$ denote the elapsed time after excitation. We define the natural frequency $\omega_n = \sqrt{k/m}$ and damping ratio $\zeta = \frac{c}{2\sqrt{km}}$. For the underdamped case ($0 < \zeta < 1$), the oscillatory response can be written as:

$$d(t) = A\,e^{-\zeta\omega_n\tau}\cos(\omega_d\tau + \phi), \qquad \text{where} \quad \omega_d = \omega_n\sqrt{1 - \zeta^2}, \tag{21}$$

where $A$ and $\phi$ are determined by the state at $t = t_1$. Crucially, the amplitude envelope $\mathcal{E}(t)$ is governed by the exponential decay term:

$$\mathcal{E}(t) = Ae^{-\zeta\omega_n\tau} = A\exp\left(-\frac{c}{2m}\tau\right). \tag{22}$$

**Implication.** In the post-excitation ring-down segment ($F(t) \approx 0$), the transient envelope decays exponentially as $\mathcal{E}(t) \propto \exp(-\eta\tau)$ with the effective damping rate

$$\eta \triangleq \frac{c}{2m}. \tag{23}$$

For beat-wise transient excitation, the effective damping rate offers a direct and interpretable characterization of amplitude attenuation, while avoiding explicit parameterization of the full MSD system. Motivated by this property, Sec. 3.2 injects a bounded gain $\alpha \in (0, 1]^D$ as a damping-inspired attenuation prior, while leaving other waveform variations to be learned by the neural backbone.

## A.2. Dataset Implementation Details

*Table 7.* Summary of datasets used in this work.

| Dataset | Signals | Scale | Overall Duration |
|---|---|---|---|
| Clinical radar (Schellenberger et al., 2020) | Raw Radar, ECG | 30 subj. (14M/16F) | ∼24h |
| PTB-XL (Wagner et al., 2020) | ECG | 21,799 rec. (18,869 subj) | ∼61h |
| Self-collected | Raw Radar, ECG | 10 subj. (6M/4F) | ∼3h |
| AFib dataset (partial release (Yuan et al., 2025)) | Processed Radar, ECG | 20 subj. (60 seg.) | 10min |

**Dataset (i).** We organize dataset (i) by scenario (Resting/Valsalva/Apnea/TiltUp/TiltDown) and perform a *subject-disjoint* split to prevent identity leakage. Specifically, subjects are randomly partitioned into 80% for training and 20% for testing with a fixed seed, and the same subject split is shared across all scenarios. All signals are uniformly resampled to 200 Hz and segmented into fixed-length windows of 512 samples with stride 128. The total number of segments is approximately 132K.

For the two-stage training, we pre-train the VAEs using four scenarios (Valsalva/Apnea/TiltUp/TiltDown) and optimize the flow model on the remaining held-out scenario (Resting), while strictly preserving the subject-disjoint split.

**Dataset (ii).** We use PTB-XL as a source ECG for inference-time synthesis: given an ECG segment from PTB-XL and a target user's metadata, the model generates the corresponding radar signal conditioned on the metadata. To ensure consistency, all PTB-XL ECG signals are resampled to 200 Hz and processed using the same normalization pipeline as dataset (i) (z-score), followed by windowing into 512-sample segments (stride 128). We synthesize approximately 23K norm segments for radar in zero-shot mmWave→ECG task and 18K segments (balanced between Normal and AFib) for AFib classification task.

**Dataset (iii).** The atrial fibrillation (AFib) dataset provides synchronized reference ECG and processed radar (a partial public disclosure with recordings from 20 subjects). For each subject, the recording is partitioned into three segments, yielding 60 labeled samples in total. We use this dataset only for downstream AFib classification evaluation, and apply the same pre-processing and segmentation pipeline (same windowing {512 samples, stride 128}) as in the other datasets. This dataset has approximately 0.9K segments which is very small dataset. The limited size of this dataset underscores the difficulty of collecting large-scale paroxysmal AFib radar data, further motivating our synthetic approach. It is only used for validation.

**Dataset (iv).** Raw mmWave measurements are first converted to range profiles. We localize the thoracic region by identifying the corresponding range bin (via peak energy within the expected chest-distance interval). We then perform beamforming to obtain directional reflections and select the stream with the highest SNR as the final radar waveform. To isolate cardiac-induced micro-vibrations, we apply a 10–40 Hz band-pass filter to the selected signal, yielding the raw radar cardiac trace. Finally, the processed radar and synchronized ECG are resampled to 200 Hz and undergo the same windowing (512 samples, stride 128) and normalization pipeline as dataset (i).

**Note:** The datasets utilized in this work originate from varying hardware configurations: Dataset (i) employs a 24GHz Continuous Wave (CW) radar, while Dataset (iv) utilizes a 60GHz Frequency Modulated Continuous Wave (FMCW) radar. Despite these differences, the fundamental sensing principles remain compatible. Physically, the carrier wavelength $\lambda$ acts primarily as a scaling coefficient for the phase response, preserving the morphological structure of the cardiac micro-vibration. Furthermore, while the modulation schemes govern the signal transmission and range resolution, they do not alter the nature of the phase-modulated echo caused by thoracic displacement. Consequently, our generative framework, which captures the underlying physiological motion dynamics, can effectively generalize across these hardware parameters.

**Our Data Acquisition Platform: Data Acquisition Platform Details.** Figure 5 illustrates our synchronized data collection system. **(Left)** The ground truth physiological signals are acquired using a CFDA-certified PC-80B ECG monitor, connected via standard wet electrodes to ensure high signal fidelity. **(Middle)** The contactless sensing frontend consists of a cascaded hardware stack: a TI IWR6843AOP (Antenna-on-Package) FMCW radar sensor (60,GHz, 3 Tx/4 Rx) mounted on an ICBOOST carrier board, which interfaces with a DCA1000EVM capture card to stream raw ADC data via Ethernet. **(Right)** The experimental setup simulates a sleep monitoring scenario. The subject lies in a supine position, with the radar suspended on an adjustable mechanical arm approximately 50,cm directly above the thoracic region. Both the radar and ECG data streams are time-synchronized and sampled at a uniform rate of 200Hz. All experiments involving human participants were approved by the Institutional Review Board (IRB) of The Hong Kong University of Science and Technology (Guangzhou) under Protocol No. HREP-2023-0322.

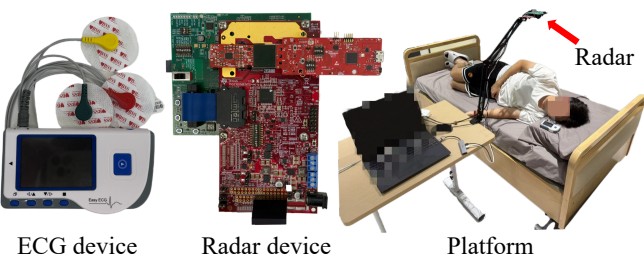

ECG device    Radar device    Platform

*Figure 5.* Synchronized radar and ECG acquisition platform.

## A.3. Sinkhorn Approximation of $\mathcal{L}_{\mathrm{WD}}$

To efficiently compute the $\mathcal{L}_{\mathcal{WD}}$, we implement a differentiable Sinkhorn solver. The procedure transforms the input latent batches into a transport cost matrix and solves for the optimal coupling via iterative dual potential updates.

---

**Algorithm 1** Differentiable Sinkhorn Calculation

---

1: **Input:** Source latents $\tilde{\mathcal{Z}}_0$, Target latents $\mathcal{Z}_1$ (Batch size $B$)
2: **Hyperparameters:** $\epsilon = 0.1$, Iterations $T = 50$, $\xi = 10^{-8}$
3: **Output:** Wasserstein estimate $\mathcal{L}_{\mathrm{WD}}$
4: *// 1. Metric Space Embedding*
5: Flatten latents: $\mathbf{X} \leftarrow \mathrm{vec}(\tilde{\mathcal{Z}}_0), \quad \mathbf{Y} \leftarrow \mathrm{vec}(\mathcal{Z}_1) \quad \in \mathbb{R}^{B \times D}$
6: Compute pairwise squared Euclidean cost $M \in \mathbb{R}^{B \times B}$:
7: $\quad M_{ij} \leftarrow \|\mathbf{x}_i - \mathbf{y}_j\|_2^2, \quad \forall i, j \in \{1, \dots, B\}$
8: *// 2. Kernel Construction with Stability*
9: Normalize cost: $\tilde{M} \leftarrow M/(\max(M) + \xi)$
10: Compute Gibbs Kernel: $K \leftarrow \exp(-\tilde{M}/\epsilon)$
11: *// 3. Sinkhorn Iterations (Dual Potentials)*
12: Initialize potentials: $\mathbf{u} \leftarrow \mathbf{1}_B, \quad \mathbf{v} \leftarrow \mathbf{1}_B$
13: Define marginals: $\boldsymbol{\mu} = \boldsymbol{\nu} = \frac{1}{B}\mathbf{1}_B$
14: **for** $t = 1$ **to** $T$ **do**
15: $\quad$ Update $\mathbf{v} \leftarrow \boldsymbol{\nu} \oslash (K^\top \mathbf{u} + \xi)$
16: $\quad$ Update $\mathbf{u} \leftarrow \boldsymbol{\mu} \oslash (K\mathbf{v} + \xi)$
17: **end for**
18: $\quad$ *$* \oslash$ denotes element-wise division*
19: *// 4. Efficient Loss Computation*
20: *$*$ Computes $\langle P, \tilde{M} \rangle$ without instantiating $P \in \mathbb{R}^{B \times B}$*
21: $\mathcal{L}_{\mathrm{WD}} \leftarrow \mathbf{u}^\top (K \odot \tilde{M})\mathbf{v}$
22: **return** $\mathcal{L}_{\mathrm{WD}}$

---

**Implementation Note.** The normalization in Step 9 is applied to the cost matrix $\tilde{M}$ to ensure numerical stability for the exponential Gibbs kernel $K$. Crucially, Step 21 computes the final loss without explicitly instantiating the dense transport plan matrix $P \in \mathbb{R}^{B \times B}$. We leverage the Sinkhorn factorization property, which defines the optimal plan as $P = \mathrm{diag}(\mathbf{u})K\mathrm{diag}(\mathbf{v})$, where $\mathbf{u}$ and $\mathbf{v}$ are the optimized dual potential vectors.

## A.4. Baseline Implementation

For traditional baselines like cVAE, cGAN and DiT, they use the same meta projector and same ECG Encoder structure as Cardio-mmflow. For SOTA model, we scale them to similar model size to make results comparable. As they are designed for PPG to generate ECG, their original structure may not very suitable for radar signal, therefore we applied necessary modifications. For RRDM, we remove its PPG domain-aware generator instead of cross-attention module. For hyperparameter, all models are tuned with AdamW (0.9,0.999) and learning for 300 epochs ($lr = 1e - 4$). 100 steps for DiT sampling. For SOTA model, we follow their paper's tuned hyperparams. For sampling inference, diffusion-based models (DiT, RDDM) utilized 100 steps, while flow-based models (PPGflowECG, Ours) utilized 10 steps Euler ODE solver.

## A.5. Evaluation Metrics Details

**PSD similarity.** Given a synthesized signal $x_{\mathrm{syn}}$ and its ground-truth counterpart $x_{\mathrm{real}}$ (sampled at $f_s = 200\,\mathrm{Hz}$), we compute their power spectral densities (PSDs) using the Welch method for robust spectral estimation. Both signals are first flattened into 1D sequences of length $L$, and the segment length is set adaptively as $n_{\mathrm{perseg}} = \min(256, \lfloor L/4 \rfloor)$ (when $L$ is sufficiently large). Welch's method then yields PSDs $P_{\mathrm{syn}}(f)$ and $P_{\mathrm{real}}(f)$ over the same frequency grid. To emphasize spectral *shape* rather than absolute magnitude, we normalize each PSD by its total power: $\tilde{P}(f) = P(f)/(\sum_f P(f) + \epsilon)$, where $\epsilon$ is a small constant for numerical stability. Finally, we quantify PSD similarity by the Pearson correlation coefficient between the normalized spectra $\tilde{P}_{\mathrm{syn}}$ and $\tilde{P}_{\mathrm{real}}$.

**Latent-FID (Distributional Realism).** We adapt the Fréchet Inception Distance (FID) to the radar domain to quantify the

distributional gap between real and synthesized signals. Concretely, we compute FID on features extracted by the encoder of a pre-trained and frozen Radar VAE. While we reuse the same pre-trained radar VAE, it is used *only* as a fixed evaluator at test time, where Latent-FID measures the distributional realism of the synthesized signals. Importantly, we do not rely on Latent-FID alone; all conclusions are corroborated with complementary time- and frequency-domain metrics as well as downstream task performance.

**HR-AE (Physiological Consistency).** To determine whether the synthesized signals retain accurate physiological rhythms, we compute the Heart Rate Absolute Error (HR-AE). The calculation follows a three-step process: First, we extract the amplitude envelope of the synthesized radar waveform to demodulate the cardiac signal. Second, we estimate the heart rate by calculating the intervals between each envelope, scaled by the specific sampling rate. Finally, we report the HR-AE as the absolute difference between the estimated heart rate of the synthetic signal and the ground-truth heart rate derived from the synchronized ECG.

### A.6. Downstream AFib Classification Implementation Details

The model is a 1D ResNet-18 variant optimized classification. It begins with an input stem featuring a 15-tap convolution (stride 2) and MaxPool to capture broad temporal morphology while reducing sequence length. The backbone consists of four residual stages with increasing feature depths of $\{64, 128, 256, 512\}$, employing basic Block1D units with integrated dropout ($p = 0.3$) for regularization. The network concludes with a Global Average Pooling layer and a linear classification head, with all weights initialized using the Kaiming Normal method to ensure stable convergence for deep signal representations.

### A.7. Coda Availability

Our codes are available on https://github.com/lanyangyang/Cardio-mmFlow.

### A.8. Discussion

**Future Directions.** (i) A natural extension of this work is the progression from single-lead to *multi-lead* ECG synthesis. This can be realized by conditioning the flow on multi-channel templates to generate radar signals that preserve lead-wise consistency, or by jointly modeling inter-lead correlations to enhance utility for morphology-sensitive tasks. (ii) Beyond signal dimensionality, addressing *motion and posture variability* remains a practical imperative. Real-world deployments involve body movements, and postural shifts that introduce non-stationary distortions and multipath effects not fully captured in our current supine-based protocol. Future research will explore explicitly modeling these factors potentially through auxiliary conditioning to establish robustness against realistic in-the-wild perturbations.

## B. Additional Qualitative Examples

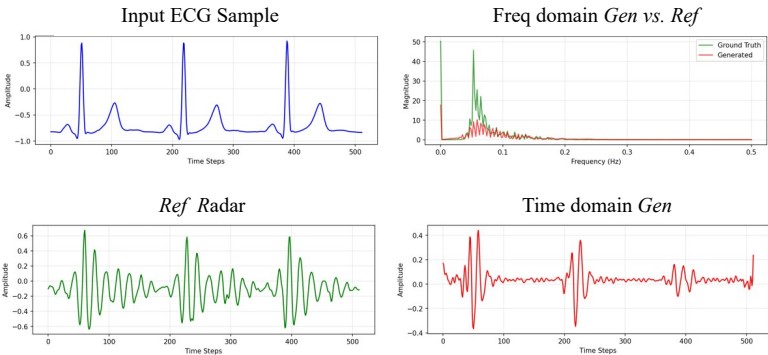

*Figure 6.* Failure samples of removing $\mathcal{L}_{freq}$ as the model only generate main frequency components but missing the small components.

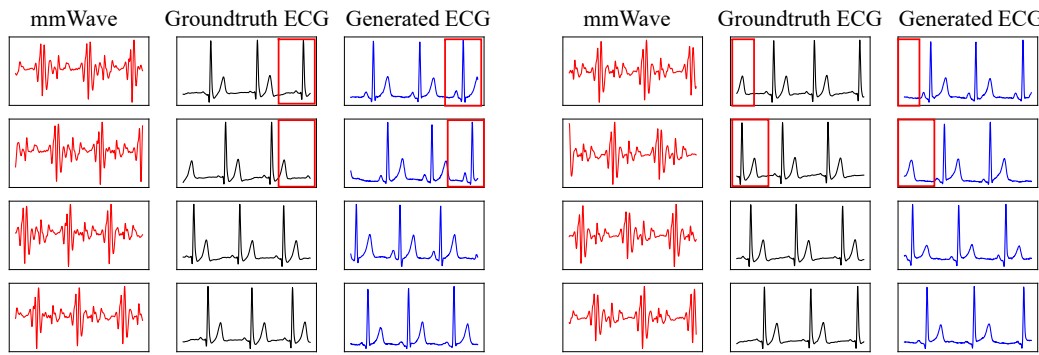

*Figure 7.* **Zero-shot downstream mmWave→ECG qualitative results.** Each row shows the input *mmWave* trace (red), the *ground-truth ECG* (black), and the *generated ECG* (blue). While most samples preserve the overall ECG morphology and rhythmic pattern in a zero-shot setting, some cases exhibit misalignment or distorted waveforms (highlighted by the red boxes), indicating residual failure modes under challenging segments. Most of them happens at the window beginning of ending where signal are often cut-off.

**Linear interpolation** in latent space from *Heart rate 58.5* to *90.1 bpm*
with subject ∈{male, Age: 40, Height: 184cm, Weight: 74kg, BMI: 21.85}

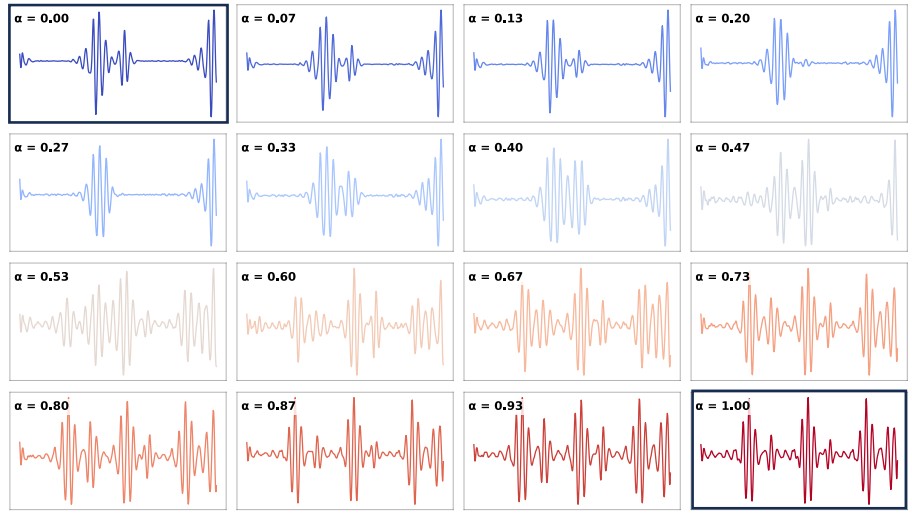

*Figure 8.* **Latent-space interpolation for controllable heart rate.** We linearly interpolate between two latent codes corresponding to heart rates from 58.5 to 90.1 bpm for a fixed subject metadata profile (male, age 40, height 184 cm, weight 74 kg, BMI 21.85). Each panel shows the synthesized waveform at interpolation coefficient $\alpha \in [0, 1]$, where $\alpha = 0$ and $\alpha = 1$ denote the two endpoints, illustrating smooth and continuous modulation of rhythm characteristics across the trajectory.

