# OpenReview forum: "Cardio-mmFlow: A Gaussian-Prior-Free Physics-Informed Flow Matching Framework for Electrocardiogram to mmWave Radar Synthesis"
_ICML.cc/2026/Conference — ICML 2026 regular_

### Official Review · Reviewer_BLSi · 2026-03-01

**Soundness:** 2
**Presentation:** 3
**Significance:** 3
**Originality:** 3
**Overall Recommendation:** 4
**Confidence:** 3

**Summary:**

This paper proposes a framework called Cardio-mmFlow for synthesizing millimeter-wave radar signals from electrocardiogram (ECG) data to address the scarcity of paired radar-ECG datasets. The method uses two pre-trained variational autoencoders (VAEs) to encode ECG and radar signals into a latent space, employs a learnable projection operator for distribution alignment via Wasserstein distance, and uses a flow matching model to transform the ECG latent space into a radar latent space. A "physically driven" MSD-FiLM module conditionally controls signal generation based on subject metadata. Experiments evaluate the synthesis quality and downstream zero-sample millimeter-wave to ECG reconstruction performance.

**Compliance With Llm Reviewing Policy:**

Affirmed.

**Final Justification:**

I have nothing further to add and i would like to matain my current score.

**Key Questions For Authors:**

1.  Could you ablate MSD-FiLM against a vanilla MLP+FiLM of equal capacity, without the softplus non-negativity or exponential decay parameterization?
2.  Could you report and quantify the independent reconstruction fidelity of the two VAEs? This seems to be a key component in ensuring the upper limit of conversion.
3.  How does performance degrade when subject metadata is partially missing or noisy? The zero-shot setting assumes accurate metadata access. If replacing metadata with population defaults or adding perturbations causes sharp degradation, the practical applicability of the personalization claim would be significantly weakened.
4.  Can you provide additional downstream clinical tasks beyond AFib classification to validate the diagnostic utility of the synthesized data?

**Limitations:**

Partially. The paper briefly mentions two future directions in Appendix A.8 (multi-lead extension and motion/posture variability), but these read more as future work than a frank acknowledgment of current limitations. Several critical limitations are left undiscussed, and societal impacts are entirely absent.

Suggested limitations to add:
*  The fundamental ill-posedness of the ECG→radar mapping: the same ECG can correspond to vastly different radar signals due to respiration, posture, multipath, and radar-body geometry, none of which are captured by the five coarse metadata features.
*  The pipeline's dependence on accurate subject metadata at inference time, which may not be reliably available in real deployments.

Suggested societal impacts to add:
Synthesized radar signals used to train clinical screening models could produce misdiagnoses if the synthetic-to-real domain gap is underestimated, posing patient safety risks.

**Strengths And Weaknesses:**

# Strengths

* The problem formulation has clear practical value. Leveraging abundant public ECG corpora to synthesize scarce mmWave radar signals for alleviating the paired-data bottleneck is a well-motivated application.

* Downstream task evaluation adds practical relevance. Beyond synthesis quality metrics alone, the paper evaluates on zero-shot from mmWave to ECG reconstruction and AFib classification, which strengthens the practical significance of the work compared to pure generative quality assessment.

* Latent-space interpolation and metadata modulation visualizations provide interpretability. Figures 3 and 4 demonstrate smooth heart-rate modulation and attribute-driven signal variation, offering evidence that the model has learned a degree of controllable generation.

# Weaknesses

* The "physics-driven" framing seems a stretch. In practice, the MSD model collapses to little more than a softplus-derived decay scalar passed through FiLM — mass and stiffness never actually factor into the computation, and damping is something the MLP quietly picks up from metadata rather than anything grounded in real physical measurement. Calling this a physics-informed design feels more like retrofitting a narrative onto what is, at its core, a metadata-conditioned FiLM layer. What makes this harder to overlook is the ablation setup in Table 4: the only baseline offered is cross-attention, which differs from FiLM in architecture to begin with. Without a fair comparison against a plain MLP+FiLM of equivalent capacity and no physical motivation whatsoever, the performance gap — whatever its size — simply cannot be pinned on the physical prior. It might just as well be the conditioning structure doing the work.



* The synthesis quality assessment metrics are too coarse to achieve the claimed clinical relevance. PSD similarity, MSE, and HR-AE are all global metrics and cannot assess the morphological fidelity of clinically significant intervals/event times (e.g., PR/QRS/QT, ST segment) after the heartbeat level or R-R peak. The lack of morphological and interval-level validation is a deficiency for a system claiming clinical application.



* The capacity of the VAE decoder may be a key bottleneck to overall synthesis quality. All synthesized signals are ultimately decoded by a Radar VAE decoder. If the decoder's reconstruction quality is inherently limited, the final output will be constrained by the decoder's capacity, regardless of the performance of the flow matching model. However, this paper neither reports the reconstruction errors of the two VAEs separately nor provides qualitative visualizations comparing the VAE reconstruction results with the original signals.



* Limited clinical coverage. This evaluation focuses only on atrial fibrillation as a single arrhythmia type, while electrocardiography is used to diagnose a wider range of cardiac diseases. Furthermore, PTB-XL provides rich multi-label annotations, which this paper utilizes to validate the method in other diagnoses, thereby better demonstrating its clinical applicability and universality.

---

> ### Author Rebuttal · Authors · 2026-03-29
>
> **For W1/Q1:** As clarified in the paper, our design is not meant to instantiate a full MSD system with explicitly measured physical parameters. Instead, it introduces a damping-inspired attenuation prior as a structured inductive bias on metadata-conditioned FiLM, rather than leaving the metadata modulation fully unconstrained.
>
> To clarify this point, we supplement a vanilla MLP-FiLM ablation. The results show that vanilla MLP-FiLM can already capture part of the metadata-conditioned effect, but it consistently degrades relative to MSD-FiLM, especially on physiological fidelity-related metrics. This indicates that the gain cannot be attributed to metadata-conditioned FiLM alone, but also comes from the damping-constrained attenuation design.
> | Metric | Ours | vanilla MLP-FiLM | Change vs. |
> |---|---:|---:|---:|
> | MSE (&darr;) | 0.027 ± 0.013 | 0.032 ± 0.014 | +17.6% |
> | PSD Similarity (&uarr;) | 0.919 ± 0.078 | 0.825 ± 0.112 | −10.2% |
> | Latent-FID (&darr;) | 14.980 | 18.273 | +22.0% |
> | HR-AE (&darr;) | 6.66 | 11.39 | +71.0% |
>
> **For W2:** We supplement interval-level validation as reported in the table below. All interval measurements are computed using NeuroKit2. The results show that the errors for major intervals remain low overall. Among them, PR duration has relatively larger error as P-wave delineation is more difficult because of its weaker and less prominent morphology. In contrast, intervals associated with more salient waveform landmarks (such as Q/R/S/T) are measured more accurately, supporting the preservation of the main cardiac temporal structure.
> | Feature | Duration (ms) |
> | :--- | :--- |
> | PR | 21.235 ± 8.936 |
> | QRS | 14.154 ± 6.333 |
> | QT | 10.385 ± 6.047 |
>
> **For W3/Q2:** The two VAEs define an important upper bound on the overall conversion quality. We therefore supplement their independent reconstruction fidelity under the same settings as in our paper. The results show that both VAEs achieve strong reconstruction performance. The radar VAE is slightly worse than the ECG VAE, likely because radar waveforms have less salient and less sharply defined morphological landmarks.
> | Model | RMSE | PCC |
> | :--- | :--- | :--- |
> | ECG VAE | 0.024 ± 0.008 | 0.996 ± 0.004 |
> | Radar VAE | 0.0393 ± 0.023 | 0.954 ± 0.041 |
>
> **For Q3:** The metadata used in our work (gender, age, height, weight, and BMI) are basic anthropometric attributes that are typically easy to obtain in both clinical and home-monitoring scenarios, so complete absence of such information is relatively uncommon in practice. More importantly, we adopt classifier-free guidance (CFG) during training, where the metadata embedding is randomly replaced by a null token with a dropout rate of 0.3, explicitly encouraging robustness to missing conditions.
>
> To explain it, we supplement a zero-shot test in which the metadata is replaced by default values at inference (same gender; age = 25; height = 172 cm; weight = 67 kg; BMI = 22.6). The results are shown in the table below. These results indicate that accurate metadata improves personalization. However, replacing subject-specific metadata with coarse defaults does not cause sharp failure. This suggests that the model retains useful robustness under missing conditions.
> | Method | RMSE | PCC | RR error (ms) |
> | :--- | :--- | :--- | :--- |
> | Method A | 0.12 ± 0.02 | 0.73 ± 0.09 | 10.72 ± 4.19 |
> | Method B | 0.11 ± 0.02 | 0.79 ± 0.06 | 8.13 ± 3.94 |
>
> **For W4/Q4:** We are currently unable to provide additional downstream clinical tasks beyond AFib.
> In the cardiac radar sensing domain, the main limitation is not the availability of ECG diagnosis labels, but the lack of public, synchronized, diagnosis-specific mmWave radar benchmarks. In our setup, PTB-XL can provide diverse ECG inputs for synthesis, but there is currently no broadly adopted public paired radar dataset covering multiple cardiac diagnostic categories with corresponding real radar ground truth.
>
> Therefore, while we could synthesize radar signals for other PTB-XL disease groups, we cannot rigorously validate their diagnostic utility without real synchronized radar counterparts. This is also one of the main motivations for our work: the field is bottlenecked not only by model design, but also by the scarcity of disease-specific paired radar data. Under the current data regime, the AFib experiment should be viewed as a proof-of-concept downstream pathology test, rather than a claim of broad clinical coverage.

---

> > ### Author Rebuttal · Reviewer_BLSi · 2026-04-01
> >
> > The detailed response provided by the authors has resolved all my concerns, and I have nothing further to add.

---

### Official Review · Reviewer_uCoV · 2026-03-02

**Soundness:** 2
**Presentation:** 1
**Significance:** 2
**Originality:** 3
**Overall Recommendation:** 3
**Confidence:** 3

**Summary:**

This paper proposes Cardio-mmFlow, a cross-modal generative framework based on flow matching for synthesizing millimeter-wave radar signals from ECG signals. The system comprises two independently pretrained VAEs (ECG-VAE and Radar-VAE), a projection module mapping ECG latents to Radar latent space, ODE-based flow matching transport, a Wasserstein distance latent alignment loss, and an MSD-FiLM module that models subject-specific radar signal attenuation through body characteristic metadata (gender, age, height, weight, BMI). The training set consists of 24 GHz CW radar data from 30 subjects, with evaluation on 10 self-collected subjects and 20 cross-hardware (60 GHz FMCW) subjects.

**Compliance With Llm Reviewing Policy:**

Affirmed.

**Final Justification:**

Thank you for your detailed rebuttal. W3/Q2 is resolved. However, I still have concerns about the remaining issues.

Methodologically, the only novel component is the element-wise latent multiplication, which, as acknowledged by the authors, serves merely as an inductive bias. The other components, such as latent alignment and the flow-matching backbone, are well-established techniques in the generative modeling literature. Labeling such a modest modification as "physics-informed" or a "physical prior" raises concerns of overclaiming.

Given the incremental methodological contribution, strong experimental design and results become essential. I understand that collecting medical data is inherently difficult, but this does not justify overclaiming: only 30 subjects with distinct metadata vectors carry a substantial risk of overfitting in the MSD-FiLM module. In fact, as the authors themselves acknowledge, "overall generative performance is also supported by the latent alignment and flow transport backbone." Therefore, the added results may simply demonstrate that performance is primarily driven by these two components rather than by MSD-FiLM's ability to capture metadata-dependent variability. The most convincing way to validate the effectiveness of MSD-FiLM in leveraging metadata would be to provide experiments with a substantially larger subject pool.

For these reasons, I can only raise my score to 3, partly in recognition of the novel application setting and partly due to the added experiment with the independent FID metric.

**Key Questions For Authors:**

Q1. Can you provide a stratified performance analysis of MSD-FiLM across different metadata subspaces (e.g., by BMI quartile, by gender)? This would directly address my main concern in W2 regarding metadata coverage, and I will reconsider my evaluation based on this.

Q2. If an independently trained encoder (rather than the frozen Radar-VAE from the pipeline) were used to compute FID, would the results remain consistent?

Q3. Does the learned attenuation \alpha in MSD-FiLM exhibit patterns consistent with physical expectations (e.g., greater attenuation for higher BMI, stronger high-frequency attenuation for thicker chest walls)? Demonstrating an interpretable relationship between learned \alpha and metadata would substantially strengthen the "physics-informed" claim.

**Limitations:**

The paper includes a limitations section acknowledging the small dataset scale and the simplified physics modeling of MSD-FiLM.

**Strengths And Weaknesses:**

## Strengths

S1. The problem formulation exhibits genuine novelty. Based on this reviewer's extensive literature survey, ECG-to-Radar cross-modal generation has virtually no precedent.

S2. The core technical challenge of the reverse direction is correctly identified. The paper accurately recognizes a fundamental asymmetry between ECG→Radar and Radar→ECG: ECG signals contain no information about the subject's body surface tissue properties or the radar propagation channel, yet these factors are major determinants of radar signal morphology. Introducing MSD-FiLM to supplement this information gap through body characteristic metadata is a well-motivated design decision.

S3. The ablation study (Table 4) provides adequate component coverage.

## Weaknesses

W1. The "physics-informed" claim is substantially overstated; the MSD modeling lacks sufficient depth.
The paper claims that MSD-FiLM is "physics-informed," arguing that the Mass-Spring-Damper model describes the propagation attenuation of cardiac mechanical vibrations through chest wall tissue. However, a genuinely physics-informed approach would explicitly embed the MSD physical equations into the model architecture. Specifically, the analytical solution of an MSD system yields an exponential decay transient response whose transfer function is governed by physical parameters such as the damping ratio ζ\zeta
ζ and natural frequency ωn\omega_n
ωn​, which in turn have well-defined physical relationships with tissue density and elastic modulus.

In practice, the paper simply maps metadata through an MLP to produce α=exp⁡(−softplus(s))∈(0,1]D\alpha = \exp(-\text{softplus}(s)) \in (0,1]^D
α=exp(−softplus(s))∈(0,1]D, then applies element-wise multiplication to the latent (FiLM gating). The constraint that "attenuation coefficients should lie in (0,1]" is trivial — any sigmoid or softmax gating automatically satisfies it without requiring MSD motivation. The paper does not employ the MSD transfer function form, does not explicitly model the relationship between damping ratio and body fat percentage or chest wall thickness, and does not leverage the frequency-dependent attenuation characteristics of MSD models. The actual implementation is therefore nearly indistinguishable from standard FiLM conditioning (Perez et al., 2018).

W2. The effective training sample size for MSD-FiLM is critically insufficient, and generalization validation is weak.
MSD-FiLM must learn a mapping from 5-dimensional metadata to D-dimensional attenuation coefficients. However, since all temporal segments from the same subject share an identical metadata vector, the effective training sample size for this mapping is only 30 (not 30 × N temporal segments).
Thirty samples provide extremely sparse coverage in a 5-dimensional space, especially considering the high collinearity among features (BMI = weight/height², gender correlates strongly with height/weight), which reduces the effective dimensionality to roughly 2–3. This means the training data likely clusters into a few regions of the metadata space (e.g., "young males," "young females"), while the model's extrapolation ability in sparse regions (e.g., high-BMI young females, low-BMI elderly males) remains entirely unvalidated.

Furthermore, zero-shot evaluation on only 10 subjects is insufficient to statistically demonstrate cross-subject generalization. While 30 subjects is a typical scale for the Radar-ECG field, prior works primarily perform reconstruction tasks (Radar→ECG) in which subject variability is implicitly encoded in the high-dimensional radar input signal. The generation direction in this paper requires the model to explicitly capture subject-specific propagation through low-dimensional metadata (5 scalars), creating a fundamentally higher demand for subject diversity.

**Suggestion**: (a) Increase the number of subjects, or (b) provide stratified analysis showing MSD-FiLM performance across different metadata subspaces (e.g., by BMI quartile or gender).

W3. The Latent-FID evaluation metric raises a circularity concern.
The paper uses Latent-FID as its primary distributional evaluation metric, computed by encoding generated radar signals through the same frozen Radar-VAE encoder that is a component of the training pipeline. This creates a circularity risk: the flow matching training process itself optimizes generated samples to match the target distribution in this encoder's latent space, so evaluating with the same encoder may naturally inflate performance.
Although Appendix A.5 includes a disclaimer stating that Latent-FID is "not relied upon alone," it nevertheless serves as the primary metric for all distributional comparisons in Tables 2–4. Using an independently trained encoder (e.g., trained on a separate data split) to compute FID would be substantially more convincing.

W4. The writing quality is insufficient, with noticeable grammatical errors.
The paper contains multiple overt grammatical errors (e.g., "our system have generate," "which revealing,"), indicating inadequate proofreading.

---

> ### Author Rebuttal · Authors · 2026-03-31
>
> **For W1:** Our claim is not that MSD-FiLM instantiates a full Mass-Spring-Damper system or explicitly solves the underlying physical equations.  As clarified in the paper and appendix, the goal is not to identify physical parameters such as mass or stiffness explicitly, but to introduce a physically motivated inductive bias. To further illustrate this, we supplement a vanilla MLP+FiLM ablation, where FiLM parameters are generated directly from metadata without the attenuation constraint. The results are as follows.
> | Metric | Ours | vanilla MLP-FiLM | Change vs. |
> |---|---:|---:|---:|
> | MSE (&darr;) | 0.027 ± 0.013 | 0.032 ± 0.014 | +17.6% |
> | PSD Similarity (&uarr;) | 0.919 ± 0.078 | 0.825 ± 0.112 | −10.2% |
> | Latent-FID (&darr;) | 14.980 | 18.273 | +22.0% |
> | HR-AE (&darr;) | 6.66 | 11.39 | +71.0% |
>
> **For W2/Q1:** We note that Cardio-mmFlow does not rely solely on MSD-FiLM. It is one component that provides structured metadata-guided modulation as a physical prior. The overall generative performance is also supported by the latent alignment and flow transport backbone. To make this clearer, we supplement a subject-level stratified evaluation over Gender and BMI.
>
> Specifically, we perform a leave-one-subject-out evaluation over all 30 subjects. In each fold, we train on 29 subjects and test on the remaining unseen subject, so that every subject is evaluated exactly once under a strictly subject-disjoint setting. Based on these subject-level results, we then group the subjects by Male/Female and by BMI subgroup, and report the corresponding stratified performance.
> The stratified results show broadly consistent performance across subgroups, without clear subgroup-specific failure.
> | | Latent FID | MSE | PSD Sim. | HR-AE |
> | :--- | :--- | :--- | :--- | :--- |
> | Male | 16.79 | 0.031 ± 0.011 | 0.901 ± 0.055 | 6.05 |
> | Female | 16.44 | 0.027 ± 0.010 | 0.877 ± 0.059 | 6.84 |
>
> | | Latent FID | MSE | PSD Sim. | HR-AE |
> | :--- | :--- | :--- | :--- | :--- |
> | 18-25 | 15.47 | 0.029 ± 0.011 | 0.887 ± 0.058 | 6.60 |
> | 25-30 | 17.24 | 0.035 ± 0.006 | 0.909 ± 0.045 | 6.84 |
> | >30 | 17.91 | 0.028 ± 0.007 | 0.873 ± 0.063 | 6.78 |
>
> **For W3/Q2:** As discussed in the paper, Latent-FID is not used in isolation, and our conclusions are supported by multiple complementary metrics and downstream results. To further address the reviewer’s circularity concern, we supplement an FID experiment using a fully independent external evaluator. Specifically, we train a CNN-based autoencoder on an external public 24 GHz radar I/Q dataset [1] and compute FID from its encoder features.
>
> This evaluator is completely decoupled from our synthesis pipeline and is not used at any stage of model training. Its encoder uses 4 Conv1D-based downsampling blocks, with a mirrored decoder. For preprocessing, we apply circle fitting, band-pass filtering, downsampling, and fixed-length windowing to the raw I/Q signals. Since FID depends on the evaluator feature space, the absolute values are not directly comparable to those in the original paper. The results are in the following table. Our method still achieves the best FID under this fully independent external evaluator.
> | Method | Our | cGAN | DiT | PPG | RDDM | cVAE |
> | :--- | :--- | :--- | :--- | :--- | :--- | :--- |
> | Latent-FID | 11.13 | 11.46 | 13.96 | 23.48 | 33.67 | 38.04 |
> > [1] Edanami, Keisuke (2021), Medical radar signal dataset, Mendeley Data, V2, doi:10.17632/6rp6wrd2pr.2
>
> **For Q3:** We would like to clarify that, as described in Sec. 3.2, MSD-FiLM is not designed to identify a single physical damping coefficient or solve the full chest-wall propagation equation. Instead, we adopt a damping-centric simplification, where a nonnegative damping intensity is computed and then mapped to α via α = exp(−damp). Under this formulation, α is a D-dimensional, flow-time-dependent attenuation gain rather than a standalone scalar physical parameter. We further note that FiLM modulation contains both a multiplicative scale and an additive shift. Therefore, interpretability should be analyzed at the level of a structured, stage-dependent attenuation pathway rather than a single coefficient.
>
> To clarify this point, we supplement a subject-level stage-wise analysis. Specifically, we extract α statistics from different flow stages and summarize their relationship with body-size-related metadata. We find that the middle and late stages show clearer systematic trends, providing additional evidence that MSD-FiLM learns a structured and partially interpretable attenuation pathway rather than arbitrary FiLM scaling.
> | | 45Kg | 55kg | 65kg | 75kg | 85kg | 95kg |
> | :--- | :--- | :--- | :--- | :--- | :--- | :--- |
> | Middle Layer | 0.90472 | 0.90343 | 0.90464 | 0.90498 | 0.90684 | 0.90707 |
> | Late Layer | 0.91015 | 0.90910 | 0.90987 | 0.91149 | 0.91166 | 0.91264 |
>
> **For W4:** We will carefully revise the manuscript and thoroughly polish the language, grammar, and overall writing quality throughout the paper.

---

> > ### Author Rebuttal · Reviewer_uCoV · 2026-04-02
> >
> > Thank you for your detailed rebuttal. W3/Q2 is resolved. However, I still have concerns about the remaining issues.
> >
> > Methodologically, the only novel component is the element-wise latent multiplication, which, as acknowledged by the authors, serves merely as an inductive bias. The other components, such as latent alignment and the flow-matching backbone, are well-established techniques in the generative modeling literature. Labeling such a modest modification as "physics-informed" or a "physical prior" raises concerns of overclaiming.
> >
> > Given the incremental methodological contribution, strong experimental design and results become essential. I understand that collecting medical data is inherently difficult, but this does not justify overclaiming: only 30 subjects with distinct metadata vectors carry a substantial risk of overfitting in the MSD-FiLM module. In fact, as the authors themselves acknowledge, "overall generative performance is also supported by the latent alignment and flow transport backbone." Therefore, the added results may simply demonstrate that performance is primarily driven by these two components rather than by MSD-FiLM's ability to capture metadata-dependent variability. The most convincing way to validate the effectiveness of MSD-FiLM in leveraging metadata would be to provide experiments with a substantially larger subject pool.
> >
> > For these reasons, I can only raise my score to 3, partly in recognition of the novel application setting and partly due to the added experiment with the independent FID metric.

---

> > > ### Author Response · Authors · 2026-04-07
> > >
> > > Thanks for your acknowledegment. We appreciate the suggestion to calibrate the wording around the physics-related description, and we are pleased to adopt more carefully.
> > >
> > > **Overfit concern.** We would like to clarify that our evaluation is not restricted to fitting a small samples cohorts in a closed setting.
> > > 1. The upstream evaluation in our paper follows a subject-independent protocol. The downstream validation is conducted on a fully separate self-collected dataset. They are from different subject groups. The evaluation covers both cross-subject upstream evaluation and downstream validation on an independent cohort.
> > > 2. To further address this concern, we additionally collecte 10 new subjects and supplement an expanded evaluation on a combined 50-subject dataset, consisting of the upstream public dataset, previously self-collected dataset, and the newly added 10 subjects.
> > > 3. We also supplemented an additional cross-dataset test in which the model is trained on the upstream dataset and evaluated solely on the self-collected dataset.
> > >   Across these additional evaluations, the performance remains at a comparable level, with no obvious changing relative to the original results. Taken together, the updated dataset now **covers 50 subjects spanning both hospital and laboratory settings.** It provides broader support for the robustness of our proposed framework.
> > > | Metric | 50-Subject | Self-collected Data |
> > > | :--- | :---: | :---: |
> > > | **Latent FID** (↓) | 16.85 | 15.56 |
> > > | **MSE** (↓) | 0.027 ± 0.01 | 0.029 ± 0.01 |
> > > | **PSD Similarity** (↑) | 0.904 ± 0.05 | 0.910 ± 0.03 |
> > >
> > > **Dataset.** We would like to explain that why choose the 30 subjects dataset. Publicly accessible paired radar–ECG benchmarks remain scarce. Most of them do not provide raw data which constrain the usage. The Schellenberger dataset has about 24h synchronized multi-lead ECG and radar signal recordings under 5 scenrios which is **the only and most diverse comprehensive public benchmark dataset**.
> > >
> > > **Novelty.** We think that the reviewer may misunderstand our novelty. We claim here that our contributions is proposing **Cardio-mmFlow**, a dedicated framework for ECG-to-mmWave cardiac waveform synthesis. In this framework, we design a Gaussian-prior-free latent flow matching model that combines Wasserstein-distance-based latent alignment. The model generate starts directly from ECG latent rather than Gaussian noise. And, we building an MSD-inspired FiLM to assist the model subject-aware modulation. To the best of our knowledge, this is **the first systematic framework for radar contactless ECG data synthesis**, designed to bridge large-scale public ECG resources with radar-domain signal generation for contactless cardiac sensing.  Our work uses ECG to synthesize corresponding radar signals with diverse characteristics. This helps alleviate the severe data scarcity in this field, where collecting synchronized ECG–radar pairs is difficult, and also provides a foundation for more general pretraining. More broadly, our framework is an important part in pushing the development of remote and home-care smart health applications where long term contactless ECG monitoring is particularly valuable. We hope our framework can serve as a practical foundation for future work on data generation, benchmarking, and downstream learning in contactless ECG sensing.

---

### Official Review · Reviewer_iMF8 · 2026-03-05

**Soundness:** 3
**Presentation:** 3
**Significance:** 2
**Originality:** 3
**Overall Recommendation:** 4
**Confidence:** 4

**Summary:**

This paper proposes Cardio-mmFlow, a physics-informed generative framework for synthesizing mmWave radar cardiac signals from clinical ECG data. The method leverages a Gaussian prior free flow matching approach that transports ECG latent representations to the radar latent space using pretrained modality-specific VAEs. To account for subject variability, the authors introduce a mass spring damper (MSD) inspired conditioning mechanism implemented via FiLM modulation driven by metadata. The framework is evaluated on radar synthesis quality, downstream zero shot mmWave->ECG reconstruction, and classification using synthetic data.

**Compliance With Llm Reviewing Policy:**

Affirmed.

**Key Questions For Authors:**

1. Can you provide stronger evidence that synthesized radar signals preserve clinically meaningful morphology (e.g. expert evaluation or morphology metrics)?

2 . How does performance change when evaluated on entirely unseen acquisition setups or physical environments?

3. You provide some failure cases in Appendix B (Figures 6 and 7) where the model misses small frequency components or misaligns waveforms. Could you elaborate on what specific characteristics of the ECG input tend to trigger these failures?

4. You mention in Appendix A.2 that the datasets utilize different hardware (24GHz CW vs. 60GHz FMCW). Did you observe any specific domain gap artifacts in the synthesized signals when cross-applying models trained on one radar type to another?

**Limitations:**

yes

**Strengths And Weaknesses:**

**Strengths:**
* The paper is generally well-structured and clearly written.
* The MSD-FiLM module is a reasonable attempt to incorporate domain knowledge about propagation variability, which is often missing in purely data-driven approaches.
* The work addresses the challenge of reducing data bottlenecks in contactless cardiac monitoring via cross-modal synthesis. This is a clinically meaningful problem and timely given growing interest in non-contact sensing and home monitoring.
* The idea of transporting ECG latents instead of starting from noise is well motivated for paired physiological signals.

**Weaknesses:**
* While metrics like PSD similarity and HR error are reported, there is limited clinical validation (e.g., waveform morphology consistency, cardiologist evaluation, or biomechanical plausibility checks).
* The paired radar datasets appear relatively small (tens of subjects). In particular, the downstream pathological evaluation (AFib classification) relies on a very small real-world dataset (20 subjects, 60 segments), which limits the strength of the clinical claims.
* The evaluation mixes datasets collected from different radar hardware (24 GHz CW radar vs. 60 GHz FMCW radar). While the authors argue that the underlying phase-modulation principle remains compatible, the potential domain gap between sensing modalities deserves further discussion.

---

> ### Author Rebuttal · Authors · 2026-03-30
>
> **For W1/Q1:** In the cardiac radar sensing domain, radar is not yet a standard clinical diagnostic modality with an established expert-reading protocol, and there is currently no widely accepted criterion for direct clinical assessment of radar waveform morphology. Therefore, rather than claiming expert-level diagnostic validation, we evaluate signal-level morphological consistency.
>
> To this end, we supplement a Pearson correlation coefficient (PCC) evaluation on the normalized Hilbert envelope between synthesized radar and paired real radar. We choose the envelope representation because it captures the coarse morphological structure and energy variation pattern of oscillatory cardiac signals more directly than raw pointwise comparison. This type of representation is also commonly used in related cardiac signal analysis such as phonocardiogram processing. The results show strong envelope-level agreement, providing additional evidence for meaningful morphological consistency, as well as the PSD and heart-rate consistency demonstrated in the paper.
> | Metric | Result |
> | :--- | :--- |
> | Avg. PCC | 0.821 |
>
> **For W2/Q2:** Data scarcity remains the primary bottleneck in this domain and is precisely the motivation of our work. In our view, the key role of our proposed model is to provide an synthesis tool that generates clean cardiac radar-phase traces to support downstram task. In our paper, we use downstream experiments only as utility probes under practically important challenges.
>
> Specifically, the AFib experiment is not meant to establish definitive clinical performance, but to test pathological potential under an especially data-scarce regime. To keep this evaluation focused, we intentionally adopt a simple ResNet-18, trained solely on synthetic AFib radar data and testing on real AFib data. Thus, the result more directly reflects whether the synthesized signals preserve transferable pathology-related discriminative information. Under this setup, the result serves as a proof-of-concept pathology test rather than a broad clinical validation.
>
> Regarding unseen acquisition setups or physical environments, our model synthesizes clean cardiac radar-phase traces rather than environment-specific sensing disturbance. Therefore, performance under unseen setups is expected to be highly downstream-dependent, since the main mismatch lies in the downstream pipeline rather than in the synthesis target itself. Importantly, the downstream evaluations in our paper are already conducted under real conditions different from the upstream paired synthesis setting, which already provides initial evidence of transfer beyond the synthesis training condition.
>
> **For Q3:** The two failure cases in Appendix B reflect different mechanisms. Figure 6 comes from the ablation without spectral fidelity loss, where the model preserves the dominant rhythm but under-represents weaker frequency components. This mainly reflects sensitivity to the training objective rather than a specific ECG morphology. In contrast, the misalignment in Figure 7 mainly appears in the zero-shot downstream setting, especially near window boundaries where the cardiac cycle is partially truncated. Based on our analysis, the key trigger is therefore not a particular ECG morphology category, but the completeness of the cardiac cycle within the input window. To further explain this, we supplement a simple masking analysis by setting part of the radar input to zero before inference. (see figure at https://anonymous.4open.science/r/figure-ocbad124/compare.png)
>
> **For W3/Q4:** We did not observe a consistent hardware-specific artifact pattern in the synthesized signals. Our view is that the main difference between 24 GHz and 60 GHz lies in sensitivity scale rather than in the underlying physiological motion being sensed. Prior radar studies [1] reported heartbeat-induced chest-wall displacement on the range of 0.2–0.5 mm, which corresponds to approximately 0.20–0.50 rad at 24 GHz and 0.50–1.26 rad at 60 GHz.  In other words, 60 GHz is more sensitive, but both radar types can still clearly observe the dominant cardiac component. In our setting, the signals are further aligned through the same preprocessing pipeline, including resampling, fixed-window segmentation, and z-score normalization, so the model mainly learns the temporal / periodic correspondence between ECG and processed radar-phase traces rather than a hardware-specific amplitude style. Moreover, evaluating on a 60 GHz FMCW setup is meaningful, as it shows that the synthesized data remain useful under a representative modern mmWave configuration rather than only within the original 24 GHz training domain.
> >[1] Ramachandran, G. & Singh, M. Three-dimensional reconstruction of cardiac displacement patterns on the chest wall during the P, QRS and T-segments of the ECG by laser speckle inteferometry. Med. Biol. Eng. Comput. 27, 525–530 (1989).

---

> > ### Author Rebuttal · Reviewer_iMF8 · 2026-04-03
> >
> > Thank you for the response. My overall assessment remains Weak Accept.

---

### Official Review · Reviewer_8VxV · 2026-03-11

**Soundness:** 3
**Presentation:** 3
**Significance:** 3
**Originality:** 3
**Overall Recommendation:** 4
**Confidence:** 1

**Summary:**

Cardio-mmFlow is a physics-informed generative framework designed to synthesize realistic millimeter-wave (mmWave) radar signals from clinical electrocardiogram (ECG) data to address the scarcity of high-quality, synchronized recordings in contactless cardiac monitoring. Departing from traditional generative models that evolve from unstructured Gaussian noise, this framework employs a "Gaussian-prior-free" flow matching approach that learns a direct transport trajectory between the latent manifolds of ECG and radar. To ensure personalized accuracy, it incorporates a simplified mass-spring-damper (MSD) physical prior that uses subject metadata—such as age, weight, and height—to model individualized signal propagation and attenuation patterns. Extensive experiments demonstrate that Cardio-mmFlow generates high-fidelity radar data that significantly improves zero-shot performance in downstream tasks, such as mmWave-to-ECG reconstruction and Atrial Fibrillation (AFib) classification, effectively bridging the gap between mechanical sensing and clinical electro-physiological standards.

**Compliance With Llm Reviewing Policy:**

Affirmed.

**Final Justification:**

I appreciate the authors' response. After the rebuttal, as I am not familiar with this line of work, I keep my original score with low confidence.

**Key Questions For Authors:**

1. How does adding a "physical prior" make the generated data better than other signals/priors? Any ablation?

2. How much "real" data is still needed for a new user to get accurate results? Is the synthetic data good enough to skip the setup process entirely, or is a brief "calibration" period still recommended for medical-grade accuracy?

**Limitations:**

yes

**Strengths And Weaknesses:**

**Strengths**

**Physics-Informed Personalization:** The model incorporates a simplified Mass-Spring-Damper (MSD) physical prior that uses subject metadata (age, height, weight) to model how cardiac vibrations propagate through different body types, leading to more accurate, individualized signal synthesis.

**Effective Zero-Shot Performance:** It significantly improves downstream tasks in a zero-shot setting—such as reconstructing ECGs from radar data or identifying Atrial Fibrillation (AFib)—by training models on synthetic data when real paired datasets are scarce.

**Weaknesses:**

**Residual Failure in Challenging Segments:** While most generated samples are high-fidelity, the model still faces "failure modes" during complex segments, such as at the beginning or end of signal windows where waveforms can become misaligned or distorted.

**Modest Gains from Real-Data Calibration:** The paper notes that even when fine-tuned with real subject data (calibration), the performance gains over the purely synthetic zero-shot approach are relatively modest, suggesting potential limitations in how effectively the model integrates limited real-world samples.

---

> ### Author Rebuttal · Authors · 2026-03-29
>
> **For W1:** We include these failure cases intentionally to provide a complete and realistic case analysis, rather than highlighting only successful examples. These cases mainly reflect challenging examples under the zero-shot mmWave→ECG reconstruction downstream setting. In particular, the most common failures occur near window boundaries, where incomplete cardiac cycles make alignment and morphology recovery more difficult. Importantly, these cases are typically localized distortions or slight temporal misalignments, rather than complete waveform collapse.
>
> **For Q1:** The key benefit of the physical prior is that it constrains how metadata modulates generation in a physically meaningful manner, rather than allowing unconstrained feature modulation. We further supplement a vanilla MLP-FiLM ablation, which directly predicts FiLM parameters from metadata while removing the MSD-inspired attenuation constraint, with all other hyperparameters kept unchanged. The results are summarized below. Compared with MSD-FiLM, vanilla MLP-FiLM only shows moderate degradation in some surface-level metrics, but it causes clear deterioration in physiological fidelity, especially in HR-AE and spectral consistency. This suggests that unconstrained FiLM can still preserve some coarse radar realism, while disturbing the underlying subject-specific physiological propagation structure.
> | Metric | Ours | vanilla MLP-FiLM | Change vs. |
> |---|---:|---:|---:|
> | MSE (&darr;) | 0.027 ± 0.013 | 0.032 ± 0.014 | +17.6% |
> | PSD Similarity (&uarr;) | 0.919 ± 0.078 | 0.825 ± 0.112 | −10.2% |
> | Latent-FID (&darr;) | 14.980 | 18.273 | +22.0% |
> | HR-AE (&darr;) | 6.66 | 11.39 | +71.0% |
>
> **For W2/Q2:** We include the calibration analysis to show a more practical point: for data-hungry downstream models, limited subject-specific calibration data may only provide modest gains. In our case, this is largely because training on synthetic data already establishes a strong zero-shot starting point, so calibration mainly serves as light personalization and refinement rather than rescue from a poor initialization.
>
> At the same time, we do not view synthetic data as a full replacement for traditional ECG at present, nor do we claim that calibration can be entirely removed, especially for high-accuracy or medical-grade use. Its role is to approximate realistic variation as closely as possible and substantially reduce the burden of paired real-data collection. In practice, a brief calibration stage remains advisable when precise results are required, while the exact amount of real data still depends on the downstream task, model sensitivity, and target reliability.

---

> > ### Author Rebuttal · Reviewer_8VxV · 2026-04-02
> >
> > thanks for your response.

---

### Decision · Program_Chairs · 2026-04-30

**Decision:**

Accept (regular)

**Comment:**

Cardio-mmFlow is a physics-informed generative framework designed to synthesize realistic millimeter-wave (mmWave) radar signals from clinical electrocardiogram (ECG) data to address the scarcity of high-quality, synchronized recordings in contactless cardiac monitoring.

Strengths.
+ The paper proposes a a good physics-informed model for mmWave to ECG conversion.
+ The paper is generally well-structured and clear
+ The problem formulation is interesting and cross-modal generation for radar to ECG seems novel.

Weaknesses.
- The paper addresses a challenging problem with limited dataset. This raises concerns about generalization (iMF8)
- Reviewers expressed concerns about the metrics being too coarse to claim clinical clinical relevance (BLSi, iMF8)

The paper received four reviews with ratings 4,4,3,4 (confidence is 1,3,3,4)
Authors submitted rebuttals and engaged in discussions with reviewers (some of whom increased their scores).

The paper offers a careful design with detailed experiments. Data scarcity in any such applications is the main bottleneck; generalization of the method to real settings remains a concern.

Overall, AC feels the strengths outweigh the weaknesses.